RESEARCH COMMUNICATION

# Size control of the inner ear via hydraulic feedback

Kishore R Mosaliganti[1†], Ian A Swinburne[1†], Chon U Chan[2†‡],
Nikolaus D Obholzer[1], Amelia A Green[1], Shreyas Tanksale[1], L Mahadevan[2,3,4,5]*,
Sean G Megason[1]*

[1]Department of Systems Biology, Harvard Medical School, Boston, United States; [2]School of Engineering and Applied Sciences, Harvard University, Cambridge, United States; [3]Department of Organismal and Evolutionary Biology, Harvard University, Cambridge, United States; [4]Department of Physics, Harvard University, Cambridge, United States; [5]Kavli Institute for NanoBio Science and Technology, Harvard University, Cambridge, United States

**\*For correspondence:**
lmahadev@g.harvard.edu (LM);
megason@hms.harvard.edu (SGM)

[†]These authors contributed equally to this work

**Present address:** [‡]Institute of Molecular and Cell Biology, Agency for Science, Technology and Research (A*STAR), Singapore, Singapore

**Competing interests:** The authors declare that no competing interests exist.

**Abstract** Animals make organs of precise size, shape, and symmetry but how developing embryos do this is largely unknown. Here, we combine quantitative imaging, physical theory, and physiological measurement of hydrostatic pressure and fluid transport in zebrafish to study size control of the developing inner ear. We find that fluid accumulation creates hydrostatic pressure in the lumen leading to stress in the epithelium and expansion of the otic vesicle. Pressure, in turn, inhibits fluid transport into the lumen. This negative feedback loop between pressure and transport allows the otic vesicle to change growth rate to control natural or experimentally-induced size variation. Spatiotemporal patterning of contractility modulates pressure-driven strain for regional tissue thinning. Our work connects molecular-driven mechanisms, such as osmotic pressure driven strain and actomyosin tension, to the regulation of tissue morphogenesis via hydraulic feedback to ensure robust control of organ size.
**Editorial note:** This article has been through an editorial process in which the authors decide how to respond to the issues raised during peer review. The Reviewing Editor's assessment is that all the issues have been addressed (see decision letter).
DOI: https://doi.org/10.7554/eLife.39596.001

## Introduction

A fundamental question in developmental biology is how different organs acquire their proper sizes, which are necessary for their healthy function. The existence of control mechanisms is evident in the consistency of organ size in the face of intrinsic noise in biological reactions such as gene expression, and in the observed recovery from size perturbations during development (*Waddington, 1959*; *Debat and Peronnet, 2013*; *Rao et al., 2002*; *Lestas et al., 2010*). However, unlike in engineered systems, where there is often a clear distinction and hierarchy between the controller and the system, in organ growth one may not have a clear hierarchy—instead there may be control mechanisms distributed across tissues and across scales. Furthermore, in developmental biology, we observe an evolved system that is not necessarily robust to all experimental perturbations that we apply when trying to understand their control networks. Consequently, it can be difficult to distinguish what is necessary for growth from what controls size.

Identifying specific mechanisms that coordinate growth—to ultimately control organ size—has been difficult because the phenomenon of growth encompasses regulatory networks that can span the molecular to organismic. Classical organ transplantation and regeneration studies in the fly (*Bryant and Levinson, 1985*; *Hariharan, 2015*), mouse (*Metcalf, 1963*; *Metcalf, 1964*), and

salamander (*Twitty and Schwind, 1931*) have indicated that both organ-autonomous and non-autonomous mechanisms control size. In his 'chalone' model, Bullough proposed growth duration to be regulated by an inhibitor of proliferation that is secreted by the growing organ and upon crossing a concentration threshold stops organ growth at the target size (*Bullough and Laurence, 1964*). Modern evidence for organ intrinsic chalones exists in myostatin for skeletal muscle, GDF11 for the nervous system, BMP3 for bone, and BMP2/4 for hair (*McPherron et al., 1997*; *Wu et al., 2003*; *Plikus et al., 2008*; *Gamer et al., 2009*). Several existing models for size control are based on global positional information regulating cell proliferation based on a morphogen gradient until final organ size is achieved (*Day and Lawrence, 2000*; *Rogulja and Irvine, 2005*; *Wartlick et al., 2011*). Other models emphasize the role of local cell-cell interactions in regulating cell proliferation or cell lineages to make tissues of the correct proportions (*García-Bellido, 2009*; *Kunche et al., 2016*). Given that cells are coupled to each other through cell-cell and cell-substrate contacts, physical constraints and tissue geometry provide tissue-level feedback. More recent models emphasize the role of tissue mechanics in regulating cell proliferation via anisotropic stresses and strain rates (*Shraiman, 2005*; *Ingber, 2005*; *Savin et al., 2011*; *Hufnagel et al., 2007*; *Behrndt et al., 2012*; *Irvine and Shraiman, 2017*; *Nelson et al., 2017*; *Pan et al., 2016*). From a molecular perspective, the insulin, Hippo (*Dupont et al., 2011*; *Legoff et al., 2013*; *Pan et al., 2016*) and TOR signaling pathways (*Colombani et al., 2003*; *Zhang et al., 2000*) have been well-established as regulators of organ size. Several studies have demonstrated that genetic mutation in these pathways is sufficient to alter organ or body size through increases in cell number, cell size, or both (*Tumaneng et al., 2012*), but the mechanisms that control size in the engineering sense (e.g. feedback of size on growth rate) are generally not known.

Most size control theories have focused on regulation of cell proliferation. Control may also arise from regulation of other parameters such as cell shape, material properties, transepithelial transport, adhesion, and the extracellular environment. In particular, fluid accumulation is a feature of developmental growth for several luminized organs including the embryonic brain (*Desmond and Jacobson, 1977*; *Lowery and Sive, 2005*), eye (*Coulombre, 1956*), gut (*Bagnat et al., 2007*), Kupffer's vesicle (*Navis et al., 2013*; *Dasgupta et al., 2018*), the inner ear (*Abbas and Whitfield, 2009*; *Hoijman et al., 2015*), and the whole mammalian embryo (*Chan et al., 2019*). Water transport across an epithelium underlies these phenomenon (*Frömter and Diamond, 1972*; *Günzel and Yu, 2013*; *Rubashkin et al., 2006*; *Fischbarg, 2010*), and for the developing brain and eye it was shown that fluid accumulation coincides with increased hydrostatic pressure (*Desmond and Jacobson, 1977*; *Coulombre, 1956*). Just as water is fundamental to the size and function of a cell's cytoplasm, the fluids filling the lumens of these organs, which are central to their development and physiological function, are fundamental components of these organs. Although specific ion transporters necessary for fluid accumulation have been identified (*Lowery and Sive, 2005*; *Bagnat et al., 2007*; *Navis et al., 2013*; *Abbas and Whitfield, 2009*), it is only very recently that we are beginning to get a glimpse of how how ion transport and transepithelial fluid flow are regulated, and their role in growth control, and much still remains to be explored .

Catch-up growth during development is the phenomenon where, after growth delay or perturbation, an organ transiently elevates its growth rate relative to other organs to get back on course. During fly development, if the growth of one imaginal disc is perturbed then a hormone, ecdysone, signals to the other imaginal discs to slow their growth such that the perturbed organ can catch-up and the animal's coordinated growth can resume (*Parker and Shingleton, 2011*). The phenomenon of catch-up growth clarified ecdysone's activity as being important for size control. Catch-up growth also occurs in vertebrates: if an infant heart or kidney is transplanted into an adult, it grows faster than the surrounding tissue to catch-up to a target size (*Dittmer et al., 1974*; *Silber, 1976*). Recently, the related phenomenon of organ symmetry has been addressed in the context of tails and the inner ear; but, the control mechanism underlying catch-growth was not clearly identified (*Das et al., 2017*; *Green et al., 2017*). Catch-up growth also occurs during bone growth and its study has clarified insulin signaling activity as being important for bone size control (*Roselló-Díez and Joyner, 2015*; *Roselló-Díez et al., 2017*; *Roselló-Díez et al., 2018*). Nonetheless, catch-up growth has been underused in the study of vertebrate-specific mechanisms of organ size control (*Roselló-Díez et al., 2018*).

Here we use a newly revealed instance of catch-up growth combined with physical theory to uncover how size control is achieved in the zebrafish otic vesicle, a fluid-filled closed epithelium that

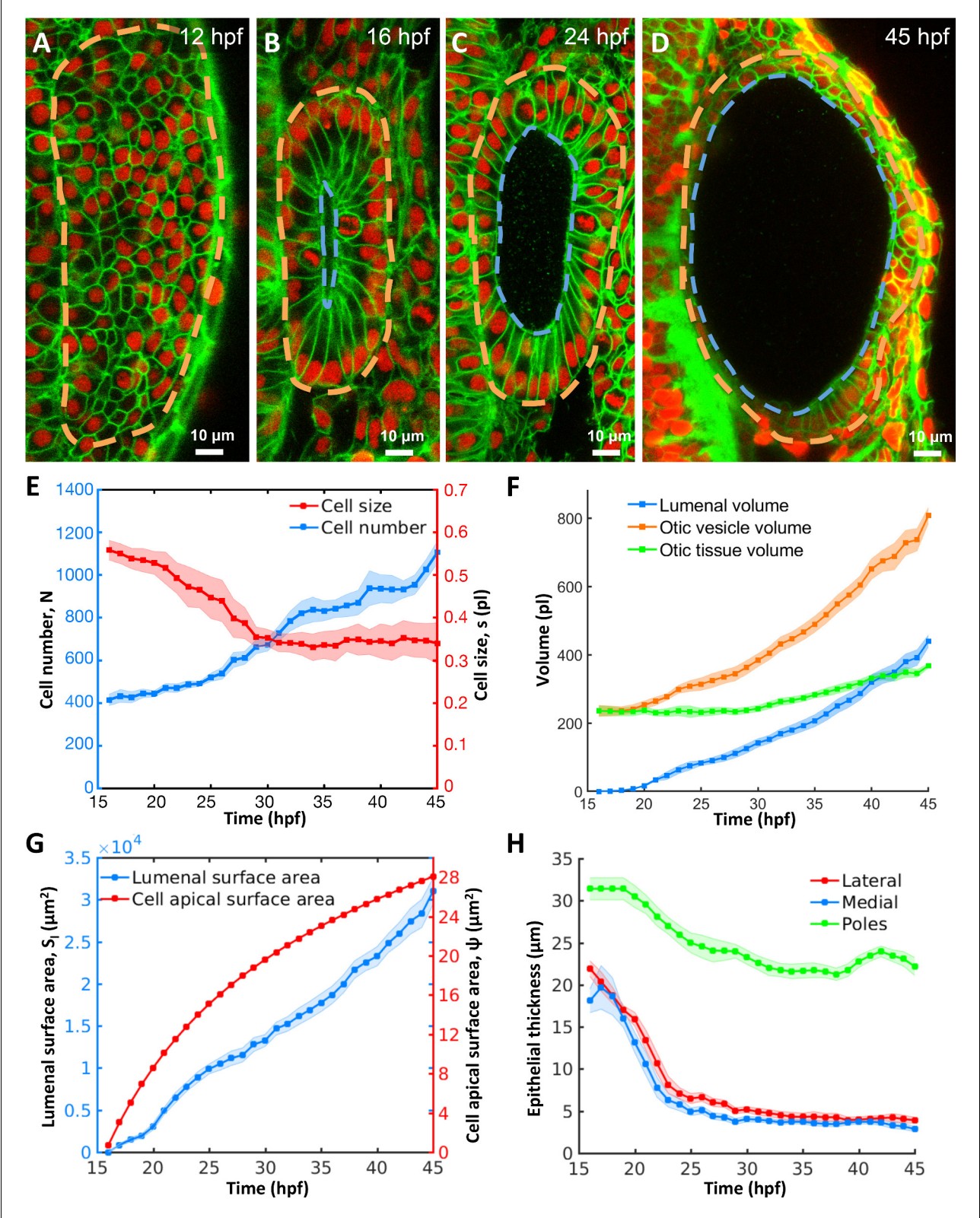

**Figure 1.** Morphodynamic analysis of inner ear growth from 16 to 45 hpf using *in toto* imaging. (A–D) Confocal micrographs of otic vesicle development at (A) 12 hpf, (B) 16 hpf, (C) 24 hpf, and (D) 45 hpf. Orange and blue contours demarcate otic vesicle and lumenal surfaces respectively. Embryos are double transgenic for highlighting membranes and nuclei (*Tg(actb2:Hsa.H2B-tdTomato)*$^{hm25}$; *Tg(actb2:mem-citrine)*$^{hm26}$). n = 10 embryos per data point. Error bars are SD. (E) Primary y-axis plots cell numbers (*N*, blue markers) and secondary axis plots average cell size (*s*, picoliters or pl,

*Figure 1 continued on next page*

*Figure 1 continued*

red markers). (**F**) Quantification of vesicle ($V_o$, orange markers), lumenal ($V_l$, blue markers), and tissue volumes ($V_t$, green markers). (**G**) Primary y-axis plots lumenal surface area ($S_l$, blue markers). Secondary axis plots average cell apical surface area ($\psi$, red markers) evaluated numerically by fitting quadratic polynomials to surface area ($S_l$) and cell number ($N$) data. (**H**) Quantification of wall thickness ($h$, μm) at locations next to the hindbrain (medial, blue), ectoderm (lateral, red), and anterioposterior poles (poles, green). Related to *Figure 1—figure supplement 1* and *Figure 1—video 1*.
DOI: https://doi.org/10.7554/eLife.39596.002

The following video and figure supplement are available for figure 1:

**Figure supplement 1.** Zebrafish inner ear growth dynamics can be quantified using *in toto* imaging protocols.
DOI: https://doi.org/10.7554/eLife.39596.003
**Figure 1—video 1.** Inflation of the otic vesicle.
DOI: https://doi.org/10.7554/eLife.39596.004

develops into the inner ear. We postulate that fluid pressure is a fundamental regulator of developmental growth in lumenized organs and hydraulic feedback can give rise to robust control of size.

## Results

### *In toto* imaging of otic vesicle development shows lumenal inflation dominates growth, not cell proliferation

We sought to determine how size control is achieved in the zebrafish otic vesicle, a 3D lumenized epithelial cyst that becomes the inner ear. Prior studies used qualitative observations and 4D imaging to examine the formation of the otic vesicle (*Haddon and Lewis, 1996*; *Hoijman et al., 2015*; *Dyballa et al., 2017*). To systematically investigate inner ear morphogenesis at longer timescales between 12–45 hours post-fertilization (hpf), we used high-resolution 3D+t confocal imaging combined with automated algorithms for quantifying cell and tissue morphology (*Figure 1—figure supplement 1A–F*) (*Megason, 2009*). Beginning at 12 hpf, bilateral regions of ectoderm adjacent to the hindbrain proliferate and subcutaneously accumulate to form the otic placodes (*Figure 1—video 1*, *Figure 1A*). The complex morphology of the inner ear arises from progressive changes in cell number, size, shape, and arrangement along with tissue-level patterns of polarization (12–14 hpf, *Figure 1A*), mesenchymal-to-epithelial transition (14–16 hpf, *Figure 1B*) and cavitation (16–24 hpf, *Figure 1C*). These steps build a closed ovoid epithelial structure, the otic vesicle, filled with a fluid called endolymph (*Figure 1—figure supplement 1G–H*). After assembly, the otic vesicle undergoes a period of rapid growth (16–45 hpf, *Figure 1D*) prior to the development of more complex substructures such as the semicircular canals and endolymphatic sac.

To evaluate growth kinetics, we used 3D image analysis (*Figure 1—figure supplement 1I–M*) to quantify a number of morphodynamic parameters between 16 and 45 hpf. During this period, cell number increased nearly three-fold from $415 \pm 26$ to $1106 \pm 52$ cells (blue curve, *Figure 1E*, for all data-points in *Figure 1* n = 10 otic vesicles, data spread is the standard deviation). However, cell proliferation was offset by a decrease in average cell size from $0.55 \pm 0.02$ pl at 16 hpf to $0.34 \pm 0.03$ pl at 28 hpf and stayed constant thereafter (red curve). Tissue volume, the product of cell number and average cell size, remained effectively constant ($230.6 \pm 7.4$ pl) until 28 hpf and subsequently increased linearly by 132 pl to 45 hpf (green curve, *Figure 1F*). The volume of the otic vesicle increased dramatically, by 572 pl from $235 \pm 16$ pl to $807 \pm 23$ pl (orange curve). The majority of the increase in size of the otic vesicle is due to an increase in lumen volume (blue curve) from 0 to $440 \pm 18$ pl (77% of the total increase) while tissue growth contributed only 23% to the increase in size.

### Pressure inflates the otic vesicle and stretches tissue viscoelastically

A mismatch between the volumetric growth of the lumen and the tissue enclosing the lumen indicates a potential role for otic tissue remodeling. Since the size of the luminal vesicle, which scales as the cube root of lumen volume, increases more rapidly than the surface area of the vesicle, which scales as the square root of the cell number enclosing that volume, we investigated how epithelial cell shape changes to accommodate growth. We observed a monotonic increase in average cell apical surface area ($\psi = S_l/N$ is lumenal surface area, $N$ is the cell number, *Figure 1G*). Since the otic

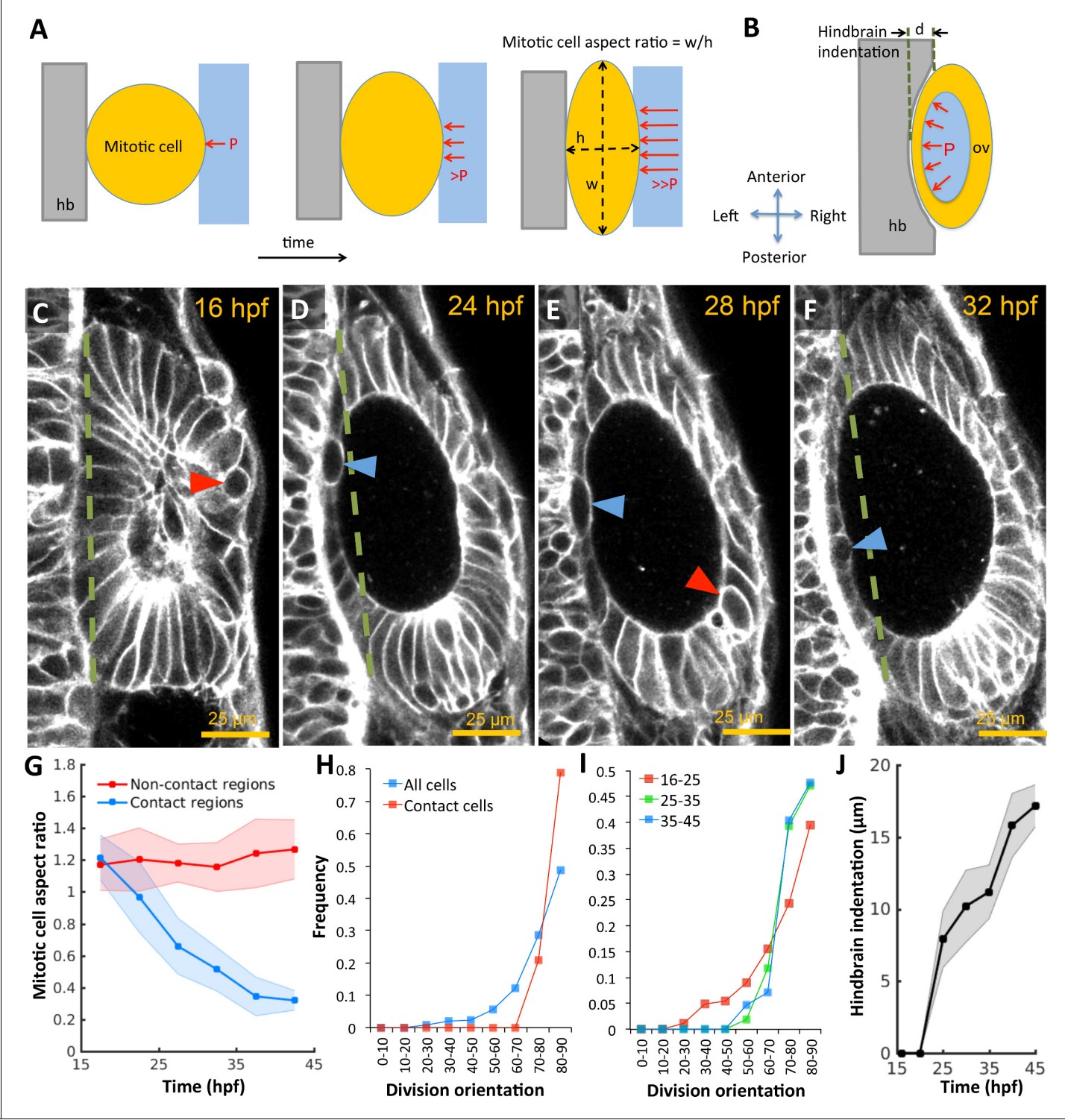

**Figure 2.** Otic vesicle growth is correlated with deformations in mitotic cell shapes and neighboring tissues that are indicative of pressure-driven strain. (**A**) Diagram illustrating inhibition of mitotic rounding just prior to cytokinesis from lumenal pressure and reactionary support from hindbrain tissue (hb, grey). (**B**) Diagram illustrating the deformation of the adjacent hindbrain tissue (hb, grey) as the otic vesicle grows from internal pressure. (**C–F**) 2D confocal micrographs of the otic vesicle at (**C**) 16 hpf, (**D**) 24 hpf, (**E**) 28 hpf, and (**F**) 32 hpf highlighting the progressive deformation of adjacent hindbrain and ectoderm tissues relative to the dashed-green line. The red and blue arrow heads highlight the progressive deformation in the shape of mitotic cells at contact and non-contact regions, respectively. (**G**) Quantification of mitotic cell aspect ratios at contact regions (hindbrain-vesicle or ectoderm-vesicle interface, blue markers) and other non-contact regions (anterioposterior poles, red markers, *n* = 54 mitotic cells total, 5–10 embryos

*Figure 2 continued on next page*

*Figure 2 continued*

per timepoint, each embryo provided 0–2 mitotic events such that each datapoint represent 4–5 mitotic events, *p<1.0e-4 at 22 hpf and *p<1.0e-5 at 27 hpf, as determined by student t-test (unpaired)). Aspect ratio is measured as the ratio of apico-basal to lateral cell radii. (H) Distribution of division plane orientation relative to the lumenal surface-normal at contact and non-contact cell populations. (I) Distribution of division plane orientation for all cells across three stages 16–25, 25–35, and 35–45 hpf respectively. (J) Quantification of hindbrain deformation measured as the peak indentation depth (relative to the dashed green line segment in C-F). *n* = 10 embryos per data point. Error bars are SD.

DOI: https://doi.org/10.7554/eLife.39596.005

epithelium is not uniform in thickness, we examined regions of the epithelium that contribute to the stretch. Except for the future sensory patches at the anterior and posterior ends (poles), epithelial thickness of the remaining otic vesicle significantly decreased from 20 µm to 4 µm during otic vesicle growth (lateral and medial regions, *Figure 1H*). The increase in lumenal volume and large cell stretching rates suggested that the vesicle is pressurized and the epithelium is under tension.

In the absence of extrinsic forces, cells round-up to a spherical morphology during mitosis by balancing internal osmotic pressure with tension provided by cortical actomyosin (*Stewart et al., 2011*). To investigate the development of pressure derived stress in the epithelium, we used mitotic cells as 'strain gauges' by measuring their deviation in aspect ratios from spheres. We observed that mitotic cells fail to round up fully in regions where the otic epithelium pushes against the hindbrain and ectoderm, in contrast to non-contact regions at the anterioposterior poles (*Figure 2A,C–G*). Furthermore, cell division planes are closely aligned with the surface-normal to the epithelium (red markers, *Figure 2H*) in comparison to the broader distribution exhibited by the non-contact cell populations (blue). The overall alignment progressively increases in developmental time as cells become more stretched (*Figure 2I*), consistent with mechanical stress driven spindle alignment previously observed in various systems including the zebrafish gastrula (*Campinho et al., 2013*), fly imaginal disc (*Legoff et al., 2013*), and zebrafish pre-enveloping layer (*Xiong et al., 2014*). Given that the otic vesicle is wedged between the hindbrain and skin (*Figure 1—figure supplement 1G–H*), we examined the impact of its volumetric growth on these tissues. We reasoned that if pressure is present, the vesicle would exhibit higher rigidity and consequently deform neighboring structures as it increased in size. To test this idea, we quantified the indentation of the hindbrain and otic vesicle interface. We observed that as the vesicle grows, the initially planar hindbrain surface indents in, and the skin bulges out (*Figure 2B–F,J*).

To directly determine the presence of pressure within the otic vesicle, we developed a novel pressure probe able to accurately measure small pressures in small volumes of liquid. This probe consists of a solid-state piezo-resistive sensor coupled to a glass capillary needle filled with water (*Figure 3A–B*, see Materials and methods). This device is capable of measuring pressure differences of 5 Pascals (≈0.5 mm of water depth) across the range of 50–400 Pascals (*Figure 3—figure supplement 1A*). Prior to 30 hpf, lumenal pressure is too low for the needle to penetrate the epithelium. From 30 hpf onwards, we observed that the needle can penetrate into the otic vesicle with no observable volume change due to leakage around the needle (*Figure 3B*). Pressure is transmitted from the otic vesicle lumen through the needle tip to the sensor. Readings after puncture increased gradually before reaching a stable pressure level (*Figure 3D*). The positive pressure remained until the glass capillary was withdrawn from the otic vesicle, after which the pressure reading dropped to the baseline value (hydrostatic pressure of the buffer due to its depth in the petri dish), further indicating that a pressure difference exists across the epithelium (*Figure 3—figure supplement 1F*). We measured the pressure level at 30, 36 and 48 hpf and found that the pressure level gradually increases from 100 Pa to upwards of 300 Pa (*Figure 3C and D*). We are uncertain whether there is a drop in pressure upon insertion of the pressure probe into the otic vesicle because there is no alternate measuring device. These values are similar to prior measurements of the much larger inner ear of adult guinea pigs (*Feldman et al., 1979*).

To directly test if lumenal pressure 'inflates' the otic vesicle to drive inner ear growth, we punctured otic vesicles at different stages between 25–45 hpf (right vesicle in *Figure 3E*). Immediately following puncture, we observed a significant decrease in vesicle diameter (white arrows, *Figure 3E*) and loss of lumenal volume (≈30–40%) (*Figure 3F*). Examination of punctured vesicles showed that as the vesicle shrunk, the epithelium became thicker (*Figure 3E and G*). Indeed, the excess surface area of the lumenal cavity was absorbed by a significant change in epithelial cell shape to become

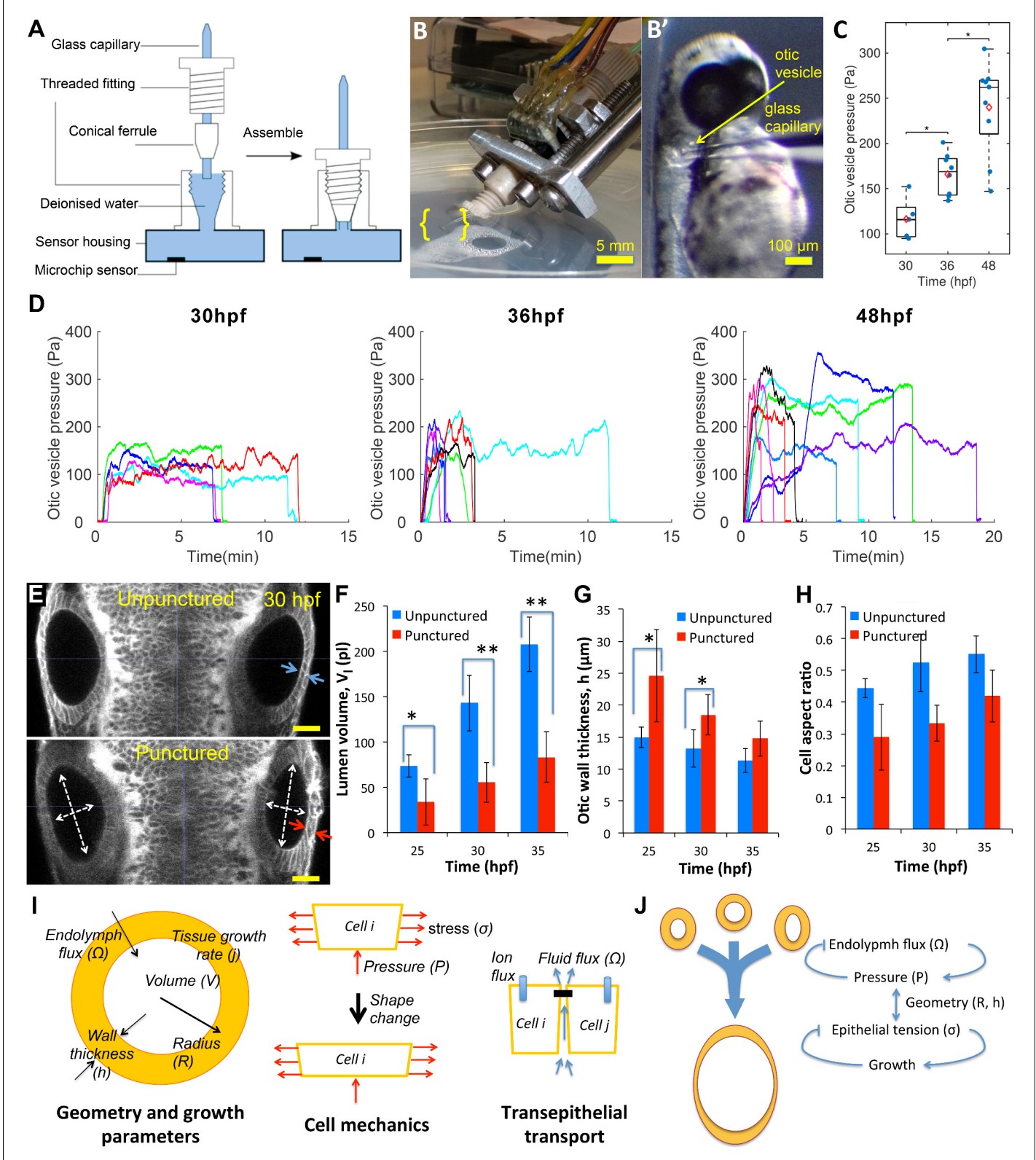

**Figure 3.** Lumenal pressure drives otic vesicle growth. Pressure measurements in the otic vesicle using a piezo-resistive solid-state sensor. (**A**) Schematic drawing of the pressure probe assembly, not to scale. (**B**) The capillary-based probe is mounted on a micromanipulator and zebrafish embryos are immobilized and mounted in Danieau buffer. (**B'**) Under a stereo microscope, the glass capillary is inserted into the otic vesicle. (**C**) Otic vesicle pressures at different developmental stages of wild-type zebrafish embryos (red diamond: mean value. *p<5.0e-2). (**D**) Pressure was measured in

*Figure 3 continued on next page*

*Figure 3 continued*

otic vesicle at 30 hpf, 36 hpf, and 48 hpf. Presented trajectories were live readings from embryos immobilized with α-bungarotoxin protein. Each color represents an individual test. (E) 2D confocal micrographs showing both ears at 30 hpf before (top) and after (bottom) unilateral puncture of the right vesicle. Changes in cell shape from squamous (blue arrows) to columnar (red arrows) are shown. Scale bar is 25 μm. (F–H) Quantification of changes from puncturing: (F) lumen volumes ($V_l$, n = 10, *p<1.0e-4,**p<1.0e-5), (G) average vesicle wall thickness ($h$, n = 10, *p<5.0e-3), and (H) average cell aspect-ratio (n = 10, error bars are SD). (I) Model relating vesicle geometry, growth rate, and fluid flux to pressure, tissue stress, and cell material properties. (J) Multi-scale regulatory control of otic vesicle growth linking pressure to fluid transport. Related to *Figure 3—figure supplements 1–2*.

DOI: https://doi.org/10.7554/eLife.39596.006

The following figure supplements are available for figure 3:

**Figure supplement 1.** Pressure probe calibration and characterizations.
DOI: https://doi.org/10.7554/eLife.39596.007
**Figure supplement 2.** Otic vesicle puncturing experiments.
DOI: https://doi.org/10.7554/eLife.39596.008

more columnar while preserving cell volume (in-plane:normal diameter change from 6.7 ± 0.2 μm:13.2 ± 2.9 μm to 5.9 ± 0.2 μm:18.4 ± 4.1 μm at 30hpf) (*Figure 3H*). A similar transition in cell shape is seen when puncturing was conducted at later stages in development (*Figure 3—figure supplement 2E*), but importantly to a less columnar resting state suggesting a viscous component. Together, the puncturing experiments provided three insights into the mechanics of the otic vesicle: (i) the lumenal fluid is under hydrostatic pressure that is released when the vesicle is punctured, (ii) lumenal pressure generates stress in the epithelium that alters the shape of epithelial cells, causing them to stretch and become flatter, and (iii) the epithelial tissue response is viscoelastic, being elastic on short time scales, consistent with the epithelium becoming thicker immediately after puncturing, and viscous at longer time scales, consistent with long-term irreversible deformations.

## Theoretical framework linking tissue geometry, fluid flux, and osmotic pressure

Given the complex interplay of lumenal pressure, geometry, and viscoelastic mechanics associated with growth, we sought to develop a mathematical model that accounts for these features (*Figure 3I*, See Materials and methods for mathematical model) (*Ruiz-Herrero et al., 2017*). In a spherically symmetric setting, the relationship between average vesicle radius ($R$), wall thickness ($h$), and tissue growth rate ($j$) can be specified as $4\pi \frac{d(R^2 h)}{dt} = j$. Similarly, the relationship between growth in lumenal volume ($V_l$) and transport across the lumenal surface (of area $S_l$) is related to fluid influx per unit surface area $\Omega = \frac{dV_l}{dt}/S_l(t)$. Defining $P_0$ as the homeostatic pressure required to balance the chemi-osmotic potential driving fluid flux and $\Omega_0$ as the flux in the absence of a pressure differential, we may write the wild-type fluid flux as $\Omega = \Omega_0 - KP_0$ where $K$ is the permeability coefficient. Intuitively, $\Omega$ is the fluid flux that maintains the homeostatic pressure $P_0$, which in turn, remodels tissue to accommodate the incoming fluid.

Changes in luminal volume can be used to directly determine fluid flux because water is incompressible at low pressures. By using population-averaged measurements of lumenal volume and surface area, we calculate that after an initial rapid expansion (16–20 hpf), flux was approximately constant ($\Omega \approx 1\mu m^3/(\mu m^2.hr) \sim 1\mu m/hr$) throughout the period 21–45 hpf (*Figure 4A*). The flux is initially high when there is no pressure but then quickly goes down as pressure builds. Thereafter, fluid accumulation can only occur through viscous expansion of the vesicle. Interestingly, analysis of the system of equations in our model shows that the vesicle will adjust endolymph flux to account for perturbations to vesicle size, via a mechanical feedback loop that links pressure to flux (*Figure 3J*). Such a control system could be useful to correct natural as well as experimentally induced asymmetry across the left-right axis as we have found in early zebrafish inner ear development (*Green et al., 2017*), and in the whole mammalian embryo (*Chan et al., 2019*).

## Model prediction and validation: pressure negatively regulates fluid flux

The model predicts that loss of fluid from the lumen (such as by puncture) should lead to a pressure drop and an increased rate of fluid flux back into the lumen and thus a higher than normal growth

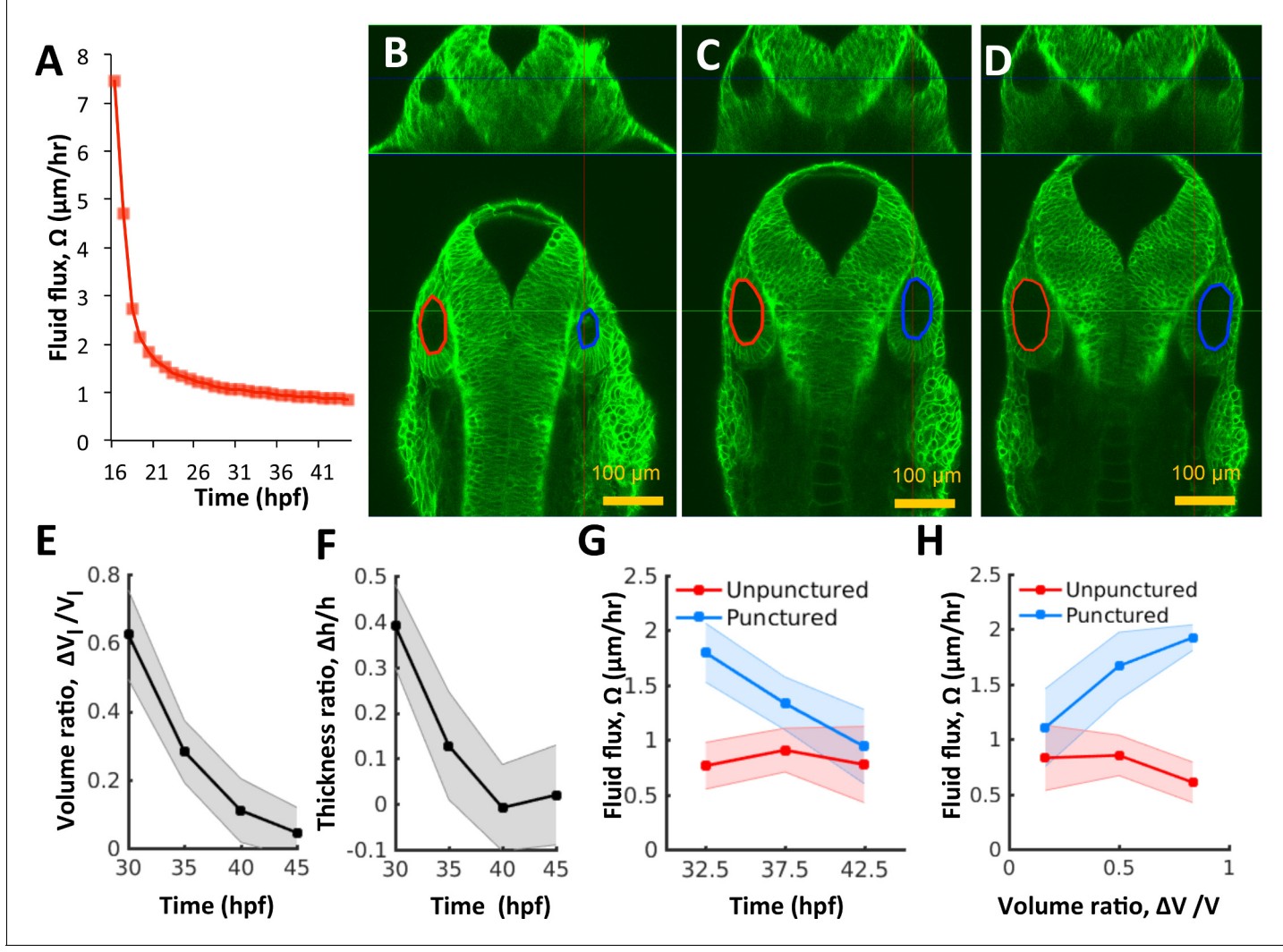

**Figure 4.** Pressure negatively regulates fluid flux. (A) Numerical calculation of fluid flux (Ω) as a function of time using *Equation 6* by fitting quadratic polynomials to volume and surface area data. (B–D) Confocal 2D micrographs with XZ (top) and XY (bottom) planes depicting the regeneration of a punctured right vesicle (blue) relative to the unpunctured vesicle (left) from (A) 30 hpf right after puncture, to (B) 32.5 hpf, and to (C) 35 hpf. (E–F) Quantification of the recovery of volume and wall thickness symmetry. The y-axis plots the difference in lumenal volumes normalized to the unpunctured lumenal volume ($\frac{\Delta V_l}{V_l}$, E) and similarly for wall thickness ($\frac{\Delta h}{h}$, F). Error bars are SD. (G) Fluid flux Ω in the punctured ears (blue) and unpunctured ears (red). Error bars are SD. (H) Scatterplot showing Ω as a function of volume asymmetry ($\frac{\Delta V_l}{V_l}$) in punctured (blue) and unpunctured (red) ears. *n* = 10 for each data point in (E–H) Related to *Figure 4—figure supplement 1* and *Figure 4—video 1*.

DOI: https://doi.org/10.7554/eLife.39596.009

The following video and figure supplement are available for figure 4:

**Figure supplement 1.** The otic vesicle regenerates to stage-specific volumes when punctured between 25–45 hpf.
DOI: https://doi.org/10.7554/eLife.39596.010

**Figure 4—video 1.** Otic vesicle catch-up growth after puncture.
DOI: https://doi.org/10.7554/eLife.39596.011

rate until size is restored, a phenomenon called catch-up growth. To test these model predictions, we experimentally examined whether pressure and fluid flux couple to each other to result in force-based feedback control of development. We first examined the response of the otic epithelium to puncturing perturbations between 25–45 hpf. We injected fluorescent dye (Alexa Fluor 594, 759 MW) into the fluid outside the inner ear (the perilymph) and tracked its movement into the lumen (*Figure 3—figure supplement 2A*). Puncturing the otic vesicle and withdrawing the needle created a loss in lumenal volume and allowed the dye from the perilymph to move into the lumen

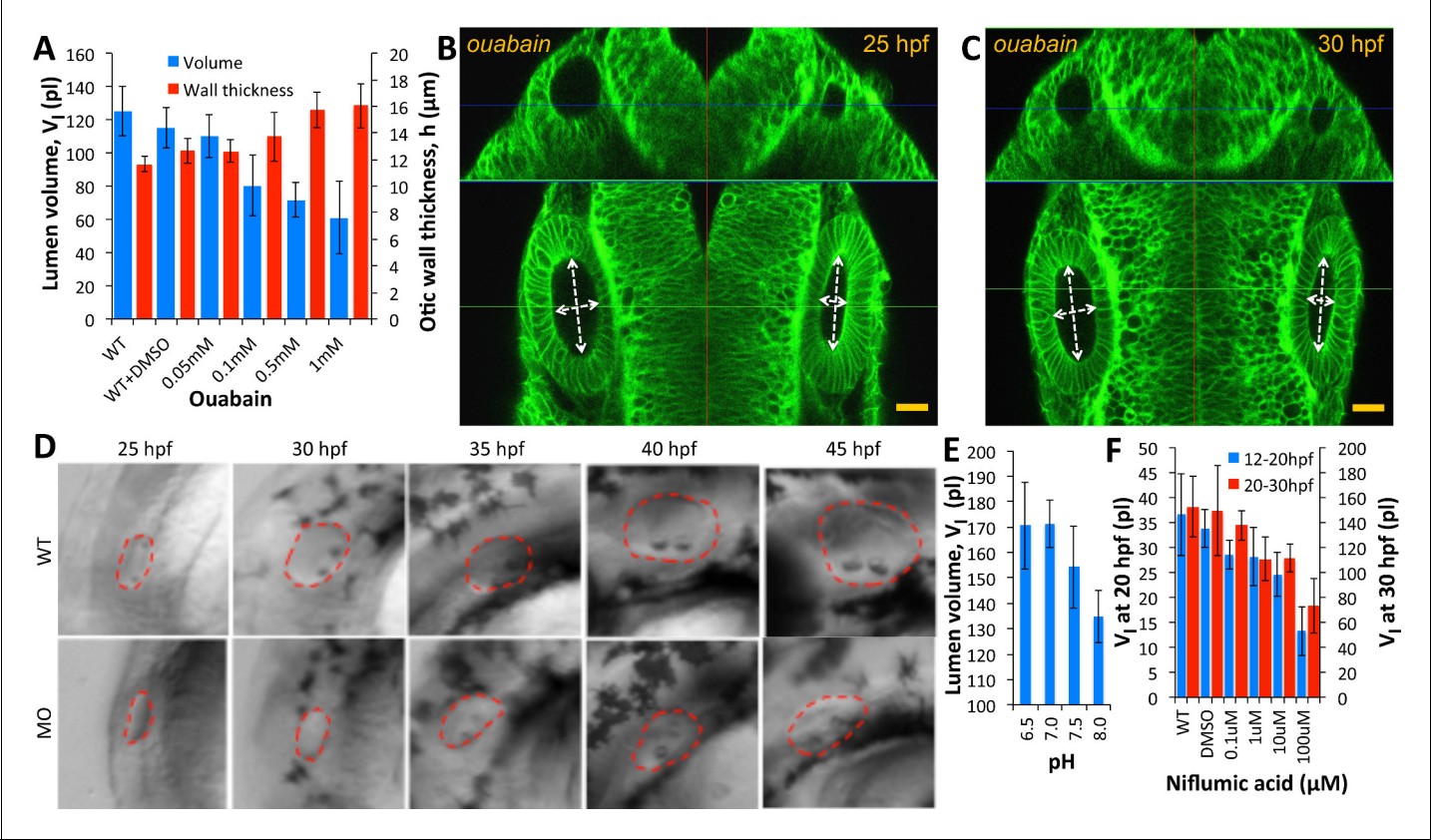

**Figure 5.** Ear size is affected by disruptions in ion transport. (**A**) Quantification of lumenal volume ($V_l$) and wall thickness ($h$) at 30 hpf after ouabain treatment at 20 hpf. Error bars are SD. (**B–C**) Confocal micrographs showing the inhibition of growth in unpunctured (left) and punctured (right) vesicles after incubation in 100 µM ouabain to the buffer at 25 hpf. Scale bar 25 µm. (**D**) Brightfield images comparing the growth (25-45hpf) of the wild-type otic vesicle against the antisense morpholino (0.25 ng) targeting the translation of Na,K-ATPase α1a.1 mRNA. (**E**) Quantification of lumenal volumes $V_l$ at 30 hpf after acidification of buffer (pH of 6.5–8.0) at 12 hpf. (**F**) Dose-dependent decrease in lumenal volumes $V_l$ observed after the addition of Niflumic acid, a chloride channel inhibitor at 12 hpf (blue) or 20 hpf (red).$n = 5$ for each data point Related to **Figure 5—video 1**.

DOI: https://doi.org/10.7554/eLife.39596.012

The following video is available for figure 5:

**Figure 5—video 1.** Abscence of both regular and catch-up growth when salt transporters inhibited.

DOI: https://doi.org/10.7554/eLife.39596.013

immediately (**Figure 3—figure supplement 2B,C**). However, when the dye was injected into the perilymph 5 min after the puncture, there was no rapid movement of dye into the lumen (**Figure 3—figure supplement 2D**). This showed that the otic epithelium rapidly seals after puncture and restores the epithelial barrier.

Next, we punctured the vesicle at 30 hpf, withdrew the needle, and evaluated its growth relative to the unpunctured contralateral vesicle (control) from 30 to 45 hpf by simultaneously imaging both otic vesicles. Interestingly, we observed the complete regeneration of lumenal volume in punctured vesicles by an increased growth rate relative to wild type to restore bilateral symmetry (**Figure 4B–E**, **Figure 4—video 1**). The cell shape changes were also reversed (**Figure 4F**) suggesting that the vesicle was re-pressurized. The rate of regeneration from fluid flux ($\Omega$) was high immediately after puncture with a slow, gradual decay as bilateral symmetry is restored (**Figure 4G**). During the rapid recovery phase, $\Omega$ in the punctured vesicle (blue curve) was 2-5X higher than that in the unpunctured vesicle (red curve). Our model predicts that upon loss of pressure $P \leq P_0$ from puncturing, the vesicle dynamically adjusts the fluid flux $\tilde{\Omega} \geq \Omega$ in linear proportion to volume lost (**Equation 13** Materials and methods). To test this prediction, we pooled data from multiple punctured embryos regenerating from varying levels of pressure loss, to measure how fluid flux related to the volume loss. Consistently, fluid flux in the punctured vesicle correlates with the difference in lumenal volumes between

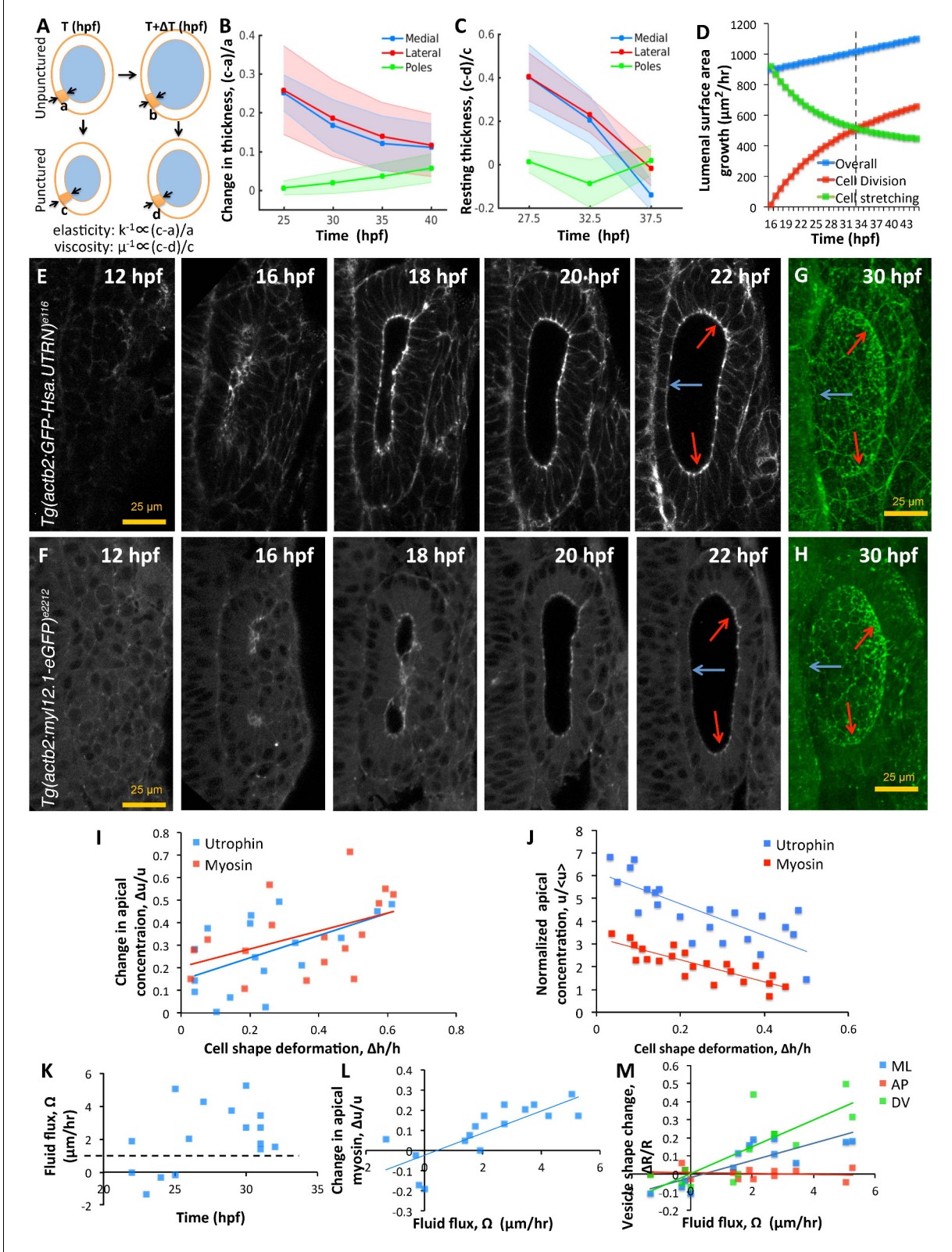

**Figure 6.** Spatial patterning of material properties results in regional thinning of tissue. (**A**) Schematic for experimentally measuring tissue material properties ($k$, $\mu$). The strain of a cell (highlighted in orange) before and after puncturing ($c - a$) is inversely proportional to the elasticity modulus k. Changes in resting cell-shapes observed over time ($d - b$) is inversely proportional to the viscosity parameter μ. (**B–C**) Quantification of normalized change in wall thickness (($c - a$)/$a$ , (**B**) and *resting* wall thickness (($c - d$)/$c$, (**C**) post-puncture near the hindbrain (medial, blue), ectoderm (lateral, red),

*Figure 6 continued on next page*

*Figure 6 continued*

and anteroposterior regions (poles, green). Puncturing was done at 25, 30, 35, and 40 hpf. *n* = 5 for each data point in (**B–C**). (**D**) Overall lumenal surface area growth rate (blue markers) showing compensatory contributions from proliferation (red) and cell stretching (green). (**E–F**) Timelapse confocal imaging using *Tg(actb2:GFP-Hsa.UTRN)* and *Tg(actb2:myl12.1-eGFP)* embryos report the dramatic apical localization of F-actin (**D**) and Myosin II (**E**) respectively prior to lumenization through 12-16hpf. Through early growth between 16–22 hpf, cells at the poles and lateral regions (red arrows) retain their fluorescence while medial cells lose their fluorescence (blue arrows). (**G–H**) 3D rendering of F-actin (**G**) and myosin II (**H**, right) data at 30 hpf show co-localization to apicolateral cell junctions as cells stretch out. (**I**) Quantification of long-term cell shape deformation ($\frac{\Delta h}{h}$) between 16–22 hpf as a function of the rate of change in apical concentration ($\frac{\Delta u}{u}$) of F-actin (blue markers) and Myosin II (red). *n* = 22. (**J**) Quantification of the short-term puncture-induced deformation in cell shapes ($\frac{\Delta h}{h}$) as a function of the normalized apical concentration ($\frac{u}{<u>}$) of F-actin (blue markers) and Myosin II (red). < u > represents the mean apical fluorescent intensity across the vesicle. Error bars are SD, *n* = 22. (**K**) Quantification of fluid flux in embryos treated with 2 mM cytochalasin D at different developmental stages (hpf). Before 25 hpf, embryos failed to grow ($\Omega \approx 0$) or lose lumenal volume. After 25 hpf, embryos increased their secretion rate by 2-5X over wild-type values (dashed black line, 1 µm/hr, *n* = 15). (**L**) Quantification showing the change in apical Myosin II fluorescence ($\frac{\Delta u}{u}$) as positively correlated with fluid flux ($\Omega$, *n* = 16). (**M**) Quantification of vesicle shape change show maximal change in dorsoventral radius (green markers) compared to the mediolateral (blue) and anteroposterior radius (red, *n* = 12). Related to *Figure 6—figure supplement 1* and *Figure 6—videos 1–4*.

DOI: https://doi.org/10.7554/eLife.39596.014

The following video and figure supplement are available for figure 6:

**Figure supplement 1.** Pole cells retain their aspect ratios as they move from high to low-curvature tissue regions between 25–30 hpf.

DOI: https://doi.org/10.7554/eLife.39596.015

**Figure 6—video 1.** Spatiotemporal dynamics of F-actin localization.

DOI: https://doi.org/10.7554/eLife.39596.016

**Figure 6—video 2.** Spatiotemporal dynamics of myosin II localization.

DOI: https://doi.org/10.7554/eLife.39596.017

**Figure 6—video 3.** Deformation of the inflated otic vesicle after treatment with cytochalasin D.

DOI: https://doi.org/10.7554/eLife.39596.018

**Figure 6—video 4.** Vesicle fails to grow upon treatment with cytochalasin D drug at 20 hpf.

DOI: https://doi.org/10.7554/eLife.39596.019

the left and right vesicles (blue markers in *Figure 4H* and *Figure 4—figure supplement 1* for other developmental stages). These data together show that vesicle pressure negatively regulates fluid flux and suggest that this feedback could buffer variations in size and drive catch-up growth in the otic vesicle.

## Ion pumps are required for lumenal expansion

The transport of salts and fluid across an epithelium can occur through a variety of mechanisms involving transcytosis, electrogenic pumps/transporters and aquaporins for transcellular or paracellular flow (*Preston et al., 1992*; *Hill and Shachar-Hill, 2006*; *Fischbarg, 2010*). Paracellular transport refers to the transfer of fluid across an epithelium by passing through the intercellular space between the cells. This is in contrast to transcellular transport, where fluid travels through the cell, passing through both the apical membrane and basolateral membrane. Previous work in the chick otic vesicle identified the activity of Na$^+$-K$^+$-ATPase in setting up a transmural potential (*Represa et al., 1986*) to drive the selective movement of water and ions. In the zebrafish, a role for ion pumps in ear growth is supported by the previous identification of the Na$^+$-K$^+$-Cl$^-$ transporter Slc12a2 as defective in *little ear* mutants (*Abbas and Whitfield, 2009*). We administered ouabain, an inhibitor of Na$^+$-K$^+$-ATPase pump activity, to embryos at the 20 hpf stage and quantified vesicle morphology at 30 hpf. We observed a dose-dependent decrease in otic vesicle volume (blue) and wall thickness deformation (red) compared to the wild-type values (*Figure 5A*) consistent with previous work (*Hoijman et al., 2015*). In punctured embryos at 25 hpf, adding 500 µM ouabain to the buffer completely inhibited further growth (left vesicle) and post-puncture regeneration (right vesicle in *Figure 5B–C*, *Figure 5—video 1*). Knockdown of Na$^+$-K$^+$-ATPase expression in morpholino-injected embryos inhibited lumenal fluid transport in a dose-dependent manner (*Figure 5D*). We additionally find that otic vesicle growth is sensitive to variations in extracellular pH and blockers of chloride channel activity (*Figure 5E–F*). Together, these data argue that a network of ion transporters for $Na^+, K^+, H^+$, and $Cl^-$ is required for fluid flux into the lumen.

## Patterning of tissue material properties causes local differences in epithelial thinning

Our minimal mathematical model assumes that the vesicle is spherical allowing us to understand and predict the qualitative trends of our experiments. For pressure $P$ acting inside a thin spherical shell, the tensional tissue-stress—the force pulling cells apart that arises from the radially-outward pushing force of hydrostatic pressure—is $\sigma = \frac{PR}{2h}$. Since the tissue is elastic on short timescales and viscous on long timescales, the radial strain-rate—the change in radius of the otic vesicle, ($\dot{\epsilon} = \frac{1}{R}\frac{dR}{dt}$)—may be related to $\sigma$ via the constitutive relation for a Maxwell fluid given by $\dot{\sigma} + \frac{\sigma}{\tau} = G\dot{\epsilon}$, where $G$ is the tissue shear modulus—the material property that relates force experienced to deformation—and $\tau$ is the ratio of the tissue viscosity $\mu$ and elasticity $k$.

For long timescales, we can use Stokes' law and force balance (**Equations 14, 16**, Materials and methods) to derive an effective tissue viscosity where

$$\mu = \frac{PR^2}{8h}\left(\frac{\mathrm{d}R}{\mathrm{d}t}\right)^{-1}. \tag{1}$$

Using this relationship, our morphodynamic measurements, and pressure measurements we estimate the effective viscosity of the otic vesicle tissue to be about $6.3 \pm 0.30 \times 10^6$ Pa*s from 24 to 36 hpf and then $2.2 \pm 0.13 \times 10^7$ Pa*s from 36 to 48 hpf (see Materials and methods for error propagation calculations). These values are within the range of tissue viscosities that had been experimentally measured, (**Gordon et al., 1972**), and indicate that the otic vesicle's tissue becomes more viscous through development.

Since the vesicle is not actually spherical (**Figure 1—figure supplement 1N,O**) and the epithelium is not uniform in thickness (**Figure 1H**; **Hoijman et al., 2015**), we examined whether (i) the non-spherical vesicle shape creates non-uniform stress distribution as in Laplace's law and/or (ii) non-uniform patterning of the material properties produce differential strain among cells. To test the first of these possibilities, we tracked anteroposterior pole cells (future sensory) and examined their shapes as they moved from high-curvature to low-curvature regions of the lumenal surface due to epithelial tread milling caused by regional differences in proliferation and emigration (**Figure 1—video 1**). We found that cells retained their columnar shapes independent of the underlying tissue curvature, suggesting that material property patterning may instead contribute to differential cell strain-rates (**Figure 6—figure supplement 1A,B**). To test the second possibility, we quantified spatial differences in elasticity ($k$) and viscosity ($\mu$) of the otic vesicle using puncturing experiments (**Figure 6A**). We observed that by eliminating pressure by puncture, medial and lateral cells deformed significantly more compared to the pole cells, indicating that they are softer (smaller $k$) (**Figure 6B**). Likewise, the resting shapes (post-puncture) of medial and lateral cells were more stretched out as the otic vesicle progressed in time, indicative of their lower viscosity ($\mu$, **Figure 6C**). The observation that the medial and lateral cells become more viscous during development (**Figure 6C**) agrees with the increased effective viscosity we independently derived from our model using measurements of unperturbed otic vesicle growth. Thus, puncturing perturbations to eliminate pressure forces allowed us to measure spatial differences in cell viscoelasticity.

## Cell stretching and steady proliferation contribute to tissue viscosity

Our model, calculations, and experimental measurements of cell material properties show that cells progressively become more rigid and more viscous. With diminished ability to remodel cell shape, we examined how otic vesicle growth can be sustained with lumenal pressure. To sustain the same growth rates, our model predicts that overall tissue viscoelasticity should be invariant to changes in cell material properties. While otic tissue elasticity arises from reversible cell stretching ($k$), tissue viscosity is the net result of irreversible cellular stretching ($\mu$) as well as proliferation-driven increase in tissue surface area. Thus, we speculated that cell stretching has a more significant role in early growth while proliferation plays a more important role in later stages of growth.

To test these predictions, we evaluated the growth in lumenal surface area ($dS_l/dt$) in terms of individual contributions from division ($\psi \frac{dN}{dt}$) and cell stretching ($N \frac{d\psi}{dt}$) (**Figure 6D**). Our analysis shows that lumenal surface area growth is linear through time (blue markers). To support this growth, the contribution from cell-stretching is high initially but monotonically decreasing (green markers) and buffered by division (red markers). A break-even point occurs at around 33 hpf when the

contribution to tissue viscosity from cell proliferation exceeds that from cell stretching (dashed black line). Interestingly, our data also show that cell shape stabilizes by this time (*Figure 1H*). Thus, cell stretching and proliferation play complementary roles through time to sustain a uniform increase in lumenal surface area.

## Tissue material properties are patterned through actomyosin regulation

To identify how cell material properties are patterned, we examined localization patterns of F-actin and Myosin II using transgenic zebrafish (*Tg(actb2:myl12.1-eGFP)*[e2212] for visualizing myosin II distribution, and *Tg(actb2:GFP-Hsa.UTRN)*[e116] for visualizing F-actin distribution (*Behrndt et al., 2012*). Both, F-actin (*Figure 6E* and *Figure 6—video 1*) and Myosin II (*Figure 6F* and *Figure 6—video 2*) were apically localized prior to lumenization through 12–16 hpf to form a band around the cavity. Through early growth between 16–22 hpf, gradual and non-uniform changes in the apical density of these molecules are observed. By 30 hpf, we find that these proteins are localized to apicolateral junctions inside cells (*Figure 6G–H*). Expression levels are retained at pole cells but reduced in medial and lateral cells. Using the movies, we tracked individual cells to understand the relationship between cell shape change and apical marker intensity. We find that wild-type cell deformation during normal growth is positively correlated to localized accumulation of F-actin and Myosin II (*Figure 6I*). In the transgenic embryos, we used a mosaic labeling strategy for tracking cells to measure the relationship between apical localization and deformation of individual cells immediately following puncture (*Figure 6—figure supplement 1C*). We find that upon puncture, the instantaneous deformation observed in individual cells is linearly correlated with the levels of apical localization of F-actin and Myosin II suggesting that actomyosin tension sets effective tissue elasticity (*Figure 6J*). As it is unclear what contribution the neighboring tissue has to the effective material properties of the growing otic vesicle, we are unable to distinguish whether the correlation between actomyosin patterns and tissue thinning is organ autonomous or whether elastic forces from neighboring tissue are influencing these behaviors.

To further link spatial patterning of actomyosin localization with epithelial thickness, we conducted loss-of-function experiments and used our model to interpret experimental results. Upon reducing cell elasticity ($k$), our model predicts: (i) an increase in strain-rate ($\dot{\epsilon}$) to equilibrate with pressure forces, and (ii) an increase in lumenal dimensions to accomodate increased strain and secretion rates. To decrease cell elasticity, we inhibited actin dynamics by treating embryos at different stages between 16–35 hpf with 100 µM cytochalasin D and used high frame-rate imaging (one frame/s) to measure vesicle deformations. As predicted by theory, we observed an increase in $\Omega$ by a factor of 2-5X over the wild-type values (*Figure 6K* and *Figure 6—video 3*). The decrease in apical myosin fluorescence positively correlated with the increase in secretion rates (*Figure 6L* and *Figure 6—video 3*). In these embryos, the DV diameter was found to increase most, compared to LR and AP diameters that experience reactionary forces from the hindbrain and skin (*Figure 6M* and *Figure 6—video 3*). We also observed that embryos between 16–25 hpf lost volume, presumably, due to a loss in epithelial connectivity and lack of pressure needed for deformation (*Figure 6—video 4*). Together, these data show how spatial patterning of the actomyosin cytokskeleton can lead to spatially varied strain in responses to spatially uniform pressure, and thus contribute to regional differences in the otic vesicle epithelium during growth.

## Discussion

Here, we show that hydrostatic lumenal pressure develops in the zebrafish otic vesicle in response to fluid transport across the otic epithelium to drive growth. We used *in toto* imaging and newly developed quantitative image analysis tools to track changes in cell number, tissue volume, and vesicle lumen volume—which is fluid flux because water is nearly incompressible. We developed a pressure probe device that is amenable to low pressures and small volumes, which enabled us to quantify a developmental increase in hydrostatic pressure. Furthermore, we identified and characterized a new instance of catch-up growth that we leveraged to develop a theoretical framework for otic vesicle size control. With the aid of a multiscale mathematical model, we hypothesized and experimentally confirmed the presence of a hydraulic negative feedback loop between pressure and fluid transport for achieving size control. Modeling helped us systematically integrate the individual contributions of cell physioligical mechanisms underlying pressure and fluid flux, cell proliferation and shape, vesicle

geometry, tissue strain-rate and viscoelasticity, to show how growth of the early otic vesicle is controlled. The negative feedback architecture that we found is similar to the chalone model of size control in that the act of growth feeds back to inhibit the rate of growth. However, compared to chalones or morphogen based growth control strategies which are limited in speed by diffusion, the pressure based strategy allows nearly instant communication between different parts of a tissue via hydraulic coupling to allow for 'course corrections' to developmental trajectories. Indeed, our study is in line with the fact that hydraulic interactions are relevant to the developmental growth of many internal organs with vesicular and tubular origins (*Ruiz-Herrero et al., 2017*), and most recently the entire mammalian embryo (*Chan et al., 2019*).

## Cause of cell thinning and origin of endolymph

Prior work found that when cells enter into mitosis and round up, their neighbors are stretched and become thinner (*Hoijman et al., 2015*). This was interpreted as being the mechanism by which cells thin to increase the surface area of the otic vesicle. Here we show that the otic epithelium is fairly elastic and cells can re-thicken following loss of lumenal pressure when the vesicle is punctured (*Figure 3G,H*), so stretching by neighboring mitotic cells may be short lived. Rather, in our model we postulate that sustained cell thinning during otic vesicle development is caused by in-plane epithelial tension in response to lumenal pressure. It was also noted that cells decrease in volume during early stages of otic vesicle growth and suggested that volume lost from cells is used to inflate the lumen (*Hoijman et al., 2015*). We attribute the reduction in cell size during early otic development to be due to cell division, and note that the net tissue volume (number of cells multiplied by average cell size) is constant at this stage (*Figure 1F*). Further, our measurements show that the lumen volume continues to grow until it exceeds the tissue volume (*Figure 1F*). We thus infer that transepithelial fluid flow is the primary if not sole source of fluid accumulation in the lumen.

## Potential application of the lumen growth model to other systems

Our model for vesicle growth integrates a range of cellular behaviors including division, transport, force generation, material property patterning, and tissue thinning. By tailoring our model equations to different geometries and growth parameters, a unified mathematical framework can be realized to understand size control in hollow organs including the eyes, brain, kidneys, vasculature, and heart. The advantages of such a mesoscale model are several. First, a mesoscale model can be more easily applied to other contexts since the level of abstraction is higher making it less dependent on the specifics of the original context (fewer parameters). Second, growth kinetics and geometry parameters can be experimentally measured using *in toto* imaging approaches. New optical technologies for measuring tissue stresses in vivo using oil droplets (*Campàs et al., 2014*) and laser-ablation (*Campinho et al., 2013*; *Hoijman et al., 2015*), ionic gradients using fluorecent ionophores (*Adams and Levin, 2012*), and pressure (*Link et al., 2004*) via probes—like the one developed here—hold promise in providing reliable biophysical measurements necessary for understanding morphogenesis. Indeed, some of these very approaches have recently been taken in helping understand how the size of a mammalian embryo is controlled (*Chan et al., 2019*). And finally, the contribution of different molecular pathways in regulating model parameters can be prioritized for experimental investigation.

## Boundary conditions of the otic vesicle and our model

The otic vesicle is not growing in isolation. In the embryo, it is immediately surrounded by extracellular matrix, mesenchymal cells, skin, and the brain. Within our model, these influences are abstracted as the effective material properties of the otic vesicle tissue. In fact, they may set limits to growth where the tension within the tissue begins to increase rapidly. We are likely observing an influence of these boundary conditions when we observe the spatial patterning of actinomyosin localization and regional tissue thinning (*Figure 6*). This boundary condition may accelerate cellular and molecular feedback mechanisms that were beyond the scope of this work. For instance, the cells within the tissue may respond to elevated tension by modulating proliferation rates, which may effectively alter the material properties of the tissue and alter strain (*Halder and Johnson, 2011*; *Gudipaty et al., 2017*; *Gnedeva et al., 2017*).

## Comparison of pressure-regulation mechanisms

Our integrated approach combining quantitative imaging and theory-guided experimentation allowed us to identify a novel hydraulic-based mechanism for regulatory control of 3D vesicle growth. This mechanism enables long-range, fast, and uniform transmission of force and connects effects at multiple scales from global pressure forces to supracellular tension and cell stretching mechanics to molecular-scale actions of ion pumps and actomyosin regulation. Later, in the adult ear, tight control of inner ear fluid pressure and ionic composition is necessary to properly detect sound, balance, and body position. Pressure is also important to maintain the structural integrity of organs. Dysregulation of pressure homeostasis can give rise to diseases including hypertension in the vasculature, Ménière's disease in the inner ear, glaucoma in the eye, and hydrocephalus in the brain. Pressure homeostasis mechanisms may vary. In the inner ear, pressure is initially regulated by feedback between pressure and lumenal fluid flux. However later in development, we have found that a physical pressure relief valve is necessary for pressure homeostasis in the inner ear (*Swinburne et al., 2018*). Together, we expect that these new insights on ear development and physiology will be critical to the development of effective clinical therapies for hearing and balance disorders, and for understanding size control in closed epithelial tissues.

# Materials and methods

**Key resources table**

| Reagent type (species) or resource | Designation | Source or reference | Identifiers | Additional information |
|---|---|---|---|---|
| Strain, strain background (*Danio rerio*) | Tg(actb2:myl12.1-EGFP)e2212 | gift from CP Heisenberg's lab, PMID: 25535919 | e2212; ZFIN ID: ZDB-ALT-130108-2 | *Behrndt et al., 2012* |
| Strain, strain background (*Danio rerio*) | Tg(actb2:GFP-Hsa.UTRN)e116 | gift from CP Heisenberg's lab, PMID: 25535919 | e116; ZFIN ID: ZDB-ALT-130206-3 | *Behrndt et al., 2012* |
| Strain, strain background (*Danio rerio*) | Tg(actb2:mem-citrine-citrine)hm30 | Megason lab, PMID: 25303534 | hm30; ZFIN ID: ZDB-ALT-150209-1 | *Xiong et al., 2014* |
| Strain, strain background (*Danio rerio*) | Tg(actb2:mem-citrine)/(actb2:Hsa.H2b-tdTomato)hm32 | Megason lab, PMID: 27535432 | hm32; ZFIN ID: ZDB-ALT-161213-1 | *Aguet et al., 2016* |
| Strain, strain background (*Danio rerio*) | Tg(actb2:mem-citrine)/(actb2:Hsa.H2b-tdTomato)hm33 | Megason lab, PMID: 27535432 | hm33; ZFIN ID: ZDB-ALT-161213-2 | *Aguet et al., 2016* |
| Strain, strain background (*Danio rerio*) | Tg(actb2:mem-mcherry2)hm29 | Megason lab, PMID: 23622240 | hm29; ZFIN ID: ZDB-ALT-120625-1 | *Xiong et al., 2013* |
| Strain, strain background (*Danio rerio*) | AB | ZIRC, Eugene, OR | ZFIN ID: ZDB-GENO-960809-7 | |
| Chemical compound, drug | Dextran, Texas Red, 3000 MW | Thermo Fisher Scientific, Waltham, MA | D-3329 | |
| Chemical compound, drug | AlexaFluor 594 | Thermo Fisher Scientific, Waltham, MA | A10442 | |
| Chemical compound, drug | Oubain | Sigma Aldrich | 11018-89-6 | |
| Chemical compound, drug | Cytochlasin D | Sigma Aldrich | C8273 | |

*Continued on next page*

*Continued*

| Reagent type (species) or resource | Designation | Source or reference | Identifiers | Additional information |
|---|---|---|---|---|
| Chemical compound, drug | Niflumic acid | Sigma Alrich | 4394-00-7 | |
| Sequence-based reagent | Morpholino | Gene tools | 5'-gccttctcctcg tcccattttgctg-3' | *Blasiole et al., 2003* |
| Commercial assay or kit | mMessage mMachine T7 ULTRA kit | Thermo Fisher Scientific, Waltham, MA | AM1345 | |
| Other | Board mount pressure sensor | Honeywell | HSCDANT001PG3A5 | pressure probe |
| Other | glass capillary | World Precision Instrument | 1B100-6 | pressure probe |
| Other | NanoPort Assembly Headless, 10-32 coned for 1/16" | idex-hs.com | n-333 | pressure probe |
| Other | Sleeve- 1517 Tefzel (ETFE) Tubing, ID 0.04", OD 1/16" | idex-hs.com | 1517 | pressure probe |
| Other | Arduino uno | sparkfun.com | R3 | pressure probe |
| Sequence-based reagent | pmtb-t7-alpha-bungarotoxin | addgene, *Swinburne et al., 2015* | Addgene #69542 | |
| Software, algorithm | *In toto* image analysis toolkit (ITIAT) | Megason lab | https://wiki.med.harvard.edu/SysBio/Megason/GoFigureImage Analysis | |
| Software, algorithm | Cmake | | | |
| Software, algorithm | ITK libraries | | www.itk.org | |
| Software, algorithm | VTK libraries | | www.vtk.org | |
| Software, algorithm | GoFigure2 | Megason lab, in preparation | www.gofigure2.org | |
| Software, algorithm | Powercrust | *Amenta et al., 2001* | https://github.com/krm15/Powercrust | |
| Software, algorithm | ACME software | *Mosaliganti et al., 2012* | | |
| Software, algorithm | MATLAB (R2014A) | www.mathworks.com | | |

## Contact for reagent and resource sharing

Further information and requests for resources and reagents should be directed to and will be fulfilled by Lead Contact, Sean Megason (megason@hms.harvard.edu).

## Experimental model and subject details

Embryos were collected using natural spawning methods and the time of fertilization was recorded according to the single cell stage of each clutch. Embryos are incubated at 28°C during imaging and all other times except room temperature during injection and dechorionation steps. Staging was recorded using hours post-fertilization (hpf) as a measure and aligned to the normal table (*Kimmel et al., 1995*).

## Zebrafish strains and maintenance

The following fluorescent transgenic strains were used in this study: (i) nuclear-localized tomato and membrane-localized citrine (*Tg(actb2:Hsa.H2B-tdTomato); Tg(actb2:mem-citrine)*[hm32,33], *Tg(actb2: mem-citrine-citrine)*[hm30] (ii) membrane-localized mCherry2 (Tg(actb2:mem-mcherry2)[hm29]) (iii) *Tg (actb2:myl12.1-eGFP)*[e2212] for visualizing myosin II distribution, and (iv) *Tg(actb2:GFP-Hsa.UTRN)*[e116]

for visualizing F-actin distribution (*Aguet et al., 2016*; *Behrndt et al., 2012*; *Xiong et al., 2014*; *Xiong et al., 2013*). All fish are housed in fully equipped and regularly maintained and inspected aquarium facilities. All fish related procedures were carried out with the approval of Institutional Animal Care and Use Committee (IACUC) at Harvard University under protocol 04877. Full details of procedures are given in Extended Experimental Procedures.

## Method details

### Timelapse confocal imaging

A canyon mount was cast in 1% agarose from a Lucite-plexiglass template and filled with 1X Danieau buffer (*Figure 1—figure supplement 1A*). The composition of the Danieau buffer is 14.4 mM sodium chloride, 0.21 mM potassium chloride, 0.12 mM magnesium sulfate, 0.18 mM calcium nitrate, and 1.5 mM HEPES buffered to pH 7.6. The template created three linear-ridges of width 400 μm, depth of 1.5 mm, and length 5 mm (*Figure 1—figure supplement 1B*). Canyon-mounted embryos developed normally for at least 3 days with a consistent orientation (lateral or dorsal or dorso-lateral) and can be continuously imaged during this time. Embryos at 15 hpf stage were dechorionated using sharp tweezers (Dumont 55) and mounted dorsally or dorso-laterally (*Figure 1—figure supplement 1C,D*) into the immersed canyon mount with a stereoscope (Leica MZ12.5). Multiple embryos for concurrent imaging were mounted in arrays (*Figure 1—figure supplement 1B*). Live imaging was performed using a Zeiss 710 confocal microscope (objectives: Plan-Apochromat 20 × 1.0 NA, C-Apochromat 40 × 1.2 NA) with a home-made heating chamber maintaining 28°C. For experiments requiring the imaging of both left and right ears in an embryo simultaneously, embryos were mounted dorsally and a Plan-Apochromat 20 × 1.0 NA objective was used. The inner ear is situated closest to the embryo surface when viewed along the dorso-lateral axis. Dorso-lateral mounting permitted the imaging of the entire ear structure with the best resolution and minimizes the depth of imaging. High-resolution imaging with a C-Apochromat 40 × 1.2 NA objective facilitated the use of automated image analysis scripts for cell and lumen segmentation, and tracking the movement of fluorescent dyes. Laser wavelengths 488 nm, 514 nm, 561 nm and 594 nm lasers were used for confocal time-courses and other single Z-stacks. Embryos were immobilized by injecting 2.3 nl of 20ng/μl $\alpha$-bungarotoxin mRNA (paralytic) at the 1 cell stage for experiments requiring long-term time-lapse imaging (*Swinburne et al., 2015*).

### Wild-type growth curves

The process of anaesthetizing an embryo to prevent twitching and preparing an embryo for continuous imaging can alter wild-type growth dynamics in the long-term. Therefore, we collected single Z-stacks for separate sets of embryos (*n* = 10–15) to establish the wild-type growth curves at hourly intervals between 16–45 hpf. These sets of embryos were immobilized rapidly by soaking in 1% tricaine.

### Confocal microscope settings

Image settings vary by brightness of signal from maternal deposit. For example, (please see *Figure 1—video 1*): labels:membrane-citrine; lasers: 514 nm (20 mW, 3%); objective: C-Apochromat 40 × 1.2 NA at 1.0 zoom; pixel dwell time: 1.58 μs; pinhole size: 89 μm; line averaging: 1; image spacing: 0.2 × 0.2 μm, and 1024 × 1024 pixels per image, with an interval of 1.0 μm through Z for 80 μm and a temporal resolution of 2 min. The starting Z location for the embryo is ≈20 μm above the top of animal pole to allow sufficient space for it to stay in the field of view or sink in the agarose (*Figure 1—figure supplement 1E,F*). A total of 25 control time-lapse (covering the developmental time-period of 15–45 hpf), 450 control Z-stacks (covering the developmental time-period of 15–45 hpf), 45 perturbation-related time-lapse datasets, and 65 perturbation-related Z-stacks were collected for the current report. Embryos were screened for their health before imaging.

### Vesicular fluid pressure probe

Our pressure probe design was inspired by capillary-based pressure sensing techniques (*Hüsken et al., 1978*; *Tomos and Leigh, 1999*). A piezo-resistive solid-state pressure sensor (Honeywell, HSC series) with high resolution (≈2 Pa, 2 kHz) and minimal mechanical deformation (detailed below) was chosen as the sensing element. The sensor was coupled via a high pressure fitting to a

≈2 cm long glass capillary (World Precision Instruments) with a conical tip of 6–13 μm inner diameter (**Figure 3A**). Before they were coupled, both the capillary and sensor were filled with deionized water and the sensor was carefully degassed to ensure the entire interior volume is filled with water. Thus, the fabricated pressure probe had a water-filled dead-end cavity with the only opening at the capillary tip. A detailed fabrication procedure will be available in a separate publication. The digital output from the sensor was sampled by a developer board at 10 Hz (Arduino Uno-R3, and in-house Matlab program). We calibrated our fabricated pressure probe by measuring the hydrostatic pressure at different water depths (**Figure 3—figure supplement 1A**) that matched with the sensor calibration provided by the manufacturer. Similar tests were conducted with various capillary diameters and ionic concentration within the bath (deionized water and a solution that resembles mature endolymph) to ensure there was no additional effect (**Figure 3—figure supplement 1B**). The composition of our synthetic endolymph was 5 mM sodium chloride, 150 mM potassium chloride, 0.2 mM calcium chloride, 0.5 mM glucose, 10 mM tris, buffered to pH 7.5.

To measure the lumenal pressure in otic vesicle, zebrafish embryos were immobilized by injecting $\alpha$-bungarotoxin mRNA and dorsolaterally positioned in a canyon mount as before. The pressure probe was mounted on a micro-manipulator typically used for injection. Puncturing was done under a stereo-microscope (**Figure 3B**, **Figure 3—figure supplement 1F**): the capillary tip was first placed next to the otic vesicle to measure the reference hydrostatic pressure, and then the tip was punctured into the otic vesicle from the lateral direction. A tight sealing was indicated by the vesicle being intact while the capillary tip was inside the vesicle. The pressure profiles are shown in **Figure 3D**. We took the mean pressure value at the plateau, that is after the initial pressure rise and before any drop due to leakage (Stage III in **Figure 3—figure supplement 1F**), as the fluid pressure inside the vesicle. The data is presented in **Figure 3C** in the main text. As a negative control, we also punctured in bulk tissue regions such as in the neural tube and measured no pressure rise.

Since the sensor, glass capillary and otic vesicle form a closed volume after puncture, any deformation on the sensing element is compensated by out-flux of the luminal fluid from the vesicle. To verify that the volume change of the piezoresistive element is negligible, and therefore does not significantly reduce the luminal pressure, we calculated the elastic deformation of the sensing membrane and compare against the otic vesicle volume. We disassembled the sensor housing and measured the membrane area to be a square with edge length $L$=850 μm. The membrane thickness, $h$, is estimated to be 5-50 μm (**Ruiz et al., 2012**). The material is simplified as an isotropic silicon plate with Young's modulus $E$=163 GPa (**Chicot et al., 1996**; **Hess, 1996**; **Dolbow and Gosz, 1996**) and Poisson's ratio $\nu$=0.27 (**Hess, 1996**; **Gan et al., 1996**). The small transverse displacement, $w$, of a thin plate under an uniform transverse hydrostatic pressure, $P$, can be calculated using the Kirchhoff-Love plate theory (**Timoshenko and Woinowsky-Krieger, 1959**) $\nabla^2\nabla^2 w = P/D$, where the bending stiffness $D = 2h^2E/3(1-\nu^2)$. The boundary condition for the built-in edges satisfies $\partial w/\partial \hat{n} = 0$, where $\hat{n}$ is the in-plane normal of the edges. The transverse deformation field, $w(x,y)$, was obtained by solving the above partial differential equation with a finite element solver (Matlab, Mathwork). An analytical solution exists for the deformation at the center as (**Timoshenko and Woinowsky-Krieger, 1959** Article 44) $w_c = 1.26 \times 10^{-3}PL^4/D$, which agrees well with the numerical solution (**Figure 3—figure supplement 1C**). The corresponding volume change, $V = \iint_L w(x,y)dxdy$, was depicted in **Figure 3—figure supplement 1D**. Comparing to the volume of a 200 μm diameter sphere (dotted line in **Figure 3—figure supplement 1D**), the volume change is at least 2 orders of magnitude smaller and hence the resultant pressure drop is negligible.

We also estimated the time scale at which the endolymph is diluted by diffusive exchange. The vesicle is a spherical domain of 100 μm diameter with the initial ionic concentration, $C_0 = 200$ mM, the estimated value in the wild type endolymph. It is connected with a conical tube ($10.5^o$ full cone angle, 1.8 mm long) with variable inner diameter of 5 - 15 μm. The tube is initially filled with liquid of 0 mM, despite the actual concentration being slightly higher due to exchange with the bath (Danieau buffer, mainly composed with 14.4 mM sodium chloride). The boundary condition of all surfaces is zero flux, except for the back end of the tip being fixed at 0 mM. The temporal evolution of concentration field was obtained by numerically solving the diffusion equation , $\partial C/\partial t = D\nabla^2 C$, where $C(\vec{r},t)$ is the concentration field and $D = 1.61^{-9}$ m$^2$/s is the diffusion coefficient of sodium chloride in water (**Guggenheim, 1954**). Example solutions and the normalized average concentration in the vesicle, $C/C_0$, is shown in **Figure 3—figure supplement 1E**. At the typical rise time (around 0.5 min) of

the probing stage, (Stage II in *Figure 3—figure supplement 1F*), the concentration remains at about 70%-90% for inner diameter of 15 μm - 5 μm. We expect the impact on the pressure reading was small at this time scale. After about 5 to 12 min, the concentration drops to 10%, which may significant modify the chemical potential. Together with the imperfection in sealing, they could contribute to the fluctuation measured at the longer time scale. However, we have ignored some factors that can maintain the ionic concentration such as active transport of ions or higher viscosity in the lumen of the otic vesicle.

## Ouabain and cytochalasin D treatment

In order to inhibit the activity of $Na^+,K^+$-ATPase, embryos at the 20 hpf stage were soaked in 1% DMSO + ouabain (Sigma Aldrich, CAS 11018-89-6) across a range of concentrations from 0 to 1 mM. Ouabain was stored at 10 mM in 1% DMSO and diluted to required concentrations in 1X Danieau buffer before use. Ear size was assessed at 30 hpf as an endpoint. For long-term imaging, ouabain was added to 1% agarose mold used for mounting the embryos. To ensure consistent penetration, 2.3 nl of 0.75 mM ouabain was injected (Nanoject) into the cardiac chamber for circulation throughout the embryo. Assuming an average embryo volume of ~180 nl, this injected dose guarantees an effective concentration of 10 μM. In order to perturb the actin network in the otic vesicle, 2.3 nl of 2 mM cytochalasin D was injected into the cardiac chamber for an effective concentration of 25 μM in the ear.

## Buffer pH and niflumic acid perturbation

To study the effect of pH on otic vesicle size, embryos at the 12 hpf stage were soaked in 1X Danieau buffer titrated to different pH levels ranging from 6.5 to 8.5 at 12 hpf. We chose to use the 12 hpf to ensure that the embryo pH homeostasis was adequately perturbed before ear growth commenced. We assessed sizes at 25 and 30 hpf. In order to inhibit the activity of chloride channels and pH regulation in the embryos, embryos at the 20 hpf stage were soaked in 1% DMSO + niflumic acid (Sigma Aldrich, CAS 4394-00-7) across a range of concentrations from 0 to 1 mM. Ear size and cell shape was assessed at 30 hpf.

## Antisense morpholino injection

A total of four α1-like and two β subunit Na,K-ATPase genes have been identified in the inner ear with distinct spatiotemporal patterns of expression (*Blasiole et al., 2003*). Antisense morpholino (Gene Tools LLC; Philomath, OR) with sequence (5'-gccttctcctcgtcccattttgctg-3') targeted against the Na,K-ATPase α-subunit gene *atp1a1a.1* (*α1a.1*) was developed to knockdown expression in the early otic vesicle (*Blasiole et al., 2006*). The ability of the morpholino to act specifically to knockdown translation of only the relevant isoform of the Na,K-ATPase mRNA was previously demonstrated using an in vitro translation assay (*Blasiole et al., 2006*). To examine the role of the Na,K-ATPase in controlling ear growth, the morpholino was injected into 1 cell wild-type zebrafish embryos. Here, we report our results from using two different doses consisting of 0.25 ng in *Figure 5D*. In comparison to wild-type phenotypes, 0.5 ng morphants developed smaller otic vesicles, displayed smaller or absent otoliths, curved tails, and lagged in overall development (data not shown). Higher doses of morpholino injection (>1 ng) made embryos unhealthy prior to otic vesicle lumenization.

## Puncturing of otic vesicle

To study the development of pressure in the otic vesicle, embryos were mounted dorso-laterally in a canyon mount (1% agarose by weight) with 1X Danieau buffer and an unclipped glass needle was slowly inserted into the otic vesicle. The needle pierced the vesicle in a lateral direction. Puncturing locally affected epithelial connectivity, causing on average 1–2 otic cell and 1–2 ectodermal cell deaths. Lumenal fluid (endolymph) leaked out along the circumference of the needle and the epithelium (*Figure 3—figure supplement 2B,C*). Needles were positioned using a micromanipulator. The needle was later slowly withdrawn to study regeneration dynamics. Thereafter, the embryo was remounted in a dorsal orientation for imaging both the ears simultaneously.

## Quantifying the viscoelastic material properties

To identify the viscoelastic material properties of otic cells, we punctured vesicles and noted the deformation in cell shape. The puncturing experiment was carried out at 5 hr intervals (25, 30, 35, 40 hpf) to determine trends in material property patterning (*Figure 6A*). We used a sample size of $n = 5 - 10$ at each timepoint. The observed deformation in the shape of a cell (before vs. after puncture) located at position $x$ and time $t$ is inversely proportional to the spring constant (*Figure 6B*). The rate of change in the resting shapes (after puncture) is inversely proportional to the viscosity coefficient (*Figure 6C*).

## Mathematical model: linking geometry, growth, mechanics and regulation

To understand otic vesicle growth, we developed a compact mathematical model that links vesicle geometry, tissue mechanics, and cellular behavior. Quantitative imaging identified aspects of cell behavior including cell division, cell size, cell shape, and material properties as being relevant to the size control problem. The process of realizing a multiscale model enabled the identification of two fundamental mechanisms regulating growth: (i) We identified a negative feedback signal linking the development of hydrostatic pressure to the inhibition of lumenal fluid flux for robust control of size; and, (ii) Spatial patterning of actomyosin contractility affected tissue response to pressure forces to shape the ear. In this section, we elaborate on the development of model equations and explain how theory-guided experimentation allowed us to arrive at the correct representation of the process.

### Conservation of mass links tissue flux and fluid flux to geometry

Quantitative analysis of vesicle geometry pointed to two critically changing parameters, namely, vesicle radii and tissue wall thickness (*Figure 1—figure supplement 1N,O* and *Figure 1H*). A simple model treats the geometry of the otic vesicle as a spherical shell (*Figure 3I*) of average radius $R = (R_o + R_l)/2$ (μm) and wall thickness $h = R_o - R_l$ (μm). As notation, variable subscripts $o$, $l$, and $t$ refer to entire otic vesicle, otic lumen, and otic tissue components respectively. To account for changes in geometry parameters from growth, we represent otic tissue growth rate as $j$ ($pl/hr$) and fluid flux (per unit surface-area) as $\Omega$ (μm/hr).

Conservation of tissue mass implies the rate of change of the otic tissue volume $j = dV_t/dt$ is related to the change in tissue geometry. Since the average thickness $h \approx 12.27 \pm 0.32$ μm and the smallest otic vesicle radius $R \approx 36.2 \pm 0.92$ μm at 30 hpf, we can make the assumption that wall-thickness is relatively small compared to the average radius ($h \ll R$), so that

$$j = \frac{dV_t}{dt} = 4\pi \frac{d(R^2 h)}{dt} \tag{2}$$

We quantified the parameters $R$, $h$, and $j$ and found that average radius increased linearly (*Figure 1—figure supplement 1N,O*), wall-thickness (*Figure 1H*) reduced asymptotically, and tissue volume (*Figure 1F*) was constant initially (16–28 hpf, $j = 0$) but linearly increased thereafter (28–45 hpf, $j \simeq 230 pl/hr$). Since tissue volume is a product of cell number ($N$) and average cell-size ($s$), we further investigated the role of these cellular parameters during growth.

$$\frac{dV_t}{dt} = \frac{dN}{dt}s + N\frac{ds}{dt} \tag{3}$$

From *Figure 1E*, cell number $N$ monotonically increased non-linearly between 16–45 hpf. In contrast, average cell-size monotonically decreased till 28 hpf and stabilized to a constant value ($\simeq 0.35 pl$) thereafter. Therefore, between 16–28 hpf, the increase in cell number was offset by a decrease in cell-size (*Figure 1E*), effectively keeping tissue mass constant. In addition, between 28–45 hpf, increase in tissue mass occurred due to increase in cell number alone with a constant average cell-size.

Similar to volume, we note that lumenal surface area is the product of cell number ($N$) and average cell apical surface area ($\psi$, μm²). Therefore, we investigated their individual contribution in driving the increase in surface area.

$$S_l = \psi N \tag{4}$$

$$\frac{dS_l}{dt} = N\frac{d\psi}{dt} + \psi\frac{dN}{dt} \tag{5}$$

From *Figure 1G and E*, we find that lumenal surface area and cell number increase monotonically from 16 to 45 hpf. Average apical cell surface showed a saturating response instead. By analyzing both the terms in *Figure 6D*, we find that both terms contribute significantly to lumenal surface area growth in a temporally complementary manner. Cell stretching ($N\frac{d\psi}{dt}$) harbors a more significant role during early growth ($\leq 32$ hpf) with a slow diminishing influence. In contrast, division ($\psi\frac{dN}{dt}$) is more significant role during later growth ($> 32$ hpf). Note that this is a second role of division during growth, in addition to the earlier role in regulating tissue volume increase (*Equation 3*).

Conservation of fluid volume implies the rate of change in lumenal volume ($\frac{dV_l}{dt}$) is equal to product of surface area ($S_l$) and fluid flux $\Omega$.

$$\frac{dV_l}{dt} = S_l\Omega \tag{6}$$

For a spherical geometry, lumenal volume is $\frac{4\pi}{3}R^3$ and surface-area is $4\pi R^2$, so that

$$\frac{4\pi}{3}\frac{d(R^3)}{dt} = 4\pi R^2\Omega \tag{7}$$

$$\frac{dR}{dt} = \Omega \tag{8}$$

In other words, fluid flux is the same as the rate of change of lumenal radius. From *Figure 4A* and *Figure 1—figure supplement 1N*, we find that average radius increased linearly and lumenal fluid flux was approximately constant ($\simeq 1\mu m/hr$) between 16–45 hpf.

Thus, our analysis of growth kinetics using simple conservation laws points to the role of mechanisms involved in regulating cell division, cell volume, fluid transport, and cell shape in controlling size.

## Modeling pressure generation and feedback to fluid transport mechanisms

Since lumenal volume growth contributes the most to vesicle growth, we examined the phenomenon of fluid transport into a closed cavity (*Figure 1F*). Prevailing theories of fluid transport suggest the movement of charged ions and fluid through intercellular junctions and channels on cell surfaces (*Hill and Shachar-Hill, 2006*; *Fischbarg, 2010*). To model the phenomenon of fluid transport into a closed cavity, we denote the rate of transport of solute as $M(mol.\mu m^{-2}.h^{-1})$ and its concentration in the lumen as $c(mol.\mu m^{-3})$. Then, fluid flux required to retain this concentration is given by the relationship:

$$\Omega = \frac{M}{c} \tag{9}$$

Earlier, our analysis showed that lumenal fluid flux $\Omega$ is a constant throughout growth. Under the assumption of isotonic transport ($c$ is a constant), we inferred that lumenal solute flux $M$ is also constant throughout growth. Since solute and fluid flux are coupled, we investigated which of these parameters are regulated. Two possible scenarios exist: (i) Otic epithelium ensures a solute flux of exactly $M$ leading to a fluid flux of $\Omega = \frac{M}{c}$, or (ii) Otic epithelium restricts fluid flux to $\Omega$ on account of wall distensibility and pressure gradient, which effectively retains a net solute flux of $M = \Omega c$. Indeed, the existence and presence of lumenal pressure is evident in terms of causing shape deformation in hindbrain tissue and mitotic cells (*Figure 2*). To concretely show the existence of pressure and how it restricts fluid transport, we performed simple perturbation experiments:

1. Puncturing experiments to verify the presence of pressure by assaying the loss of lumenal volume (*Figure 3*).

2. Regeneration experiments to demonstrate that the otic vesicle is capable of increased rates of solute and fluid transport in the absence of pressure. We measured regeneration dynamics (*Figure 4*) and found that vesicles were able to increasing flux by a factor $k = 3 \times -4 \times$ the wild-type values (*Figure 4—figure supplement 1*).

Experimental outcomes suggest that pressure creates an opposing force to the osmotic potential forces to inhibit the movement of fluid, thus setting growth rates appropriate to the developmental stage. Thus, in unperturbed wild-type embryos, the observed fluid flux ($\Omega$) is a function of the difference between the osmotically-derived flux ($\Omega_0$, arising purely due the osmotic potential alone) and the deviation from a set-point pressure, that we term the homeostatic pressure in the vesicle $P_0$. Making the assumption that the functional form is linear then suggests the equation

$$\Omega = \Omega_0 - KP_0 \tag{10}$$

In punctured embryos, the loss in pressure ($P \leq P_0$) leads to a larger flux following regeneration ($\tilde{\Omega} \geq \Omega$).

$$\tilde{\Omega} = \Omega_0 - KP \tag{11}$$

that is the increase in flux relative to the unperturbed value is proportional to the loss in pressure relative to its homeostatic value, with

$$\tilde{\Omega} - \Omega = K(P_0 - P) \tag{12}$$

For small changes in the radius in the neighborhood of the fixed point, the loss-in-pressure is proportional to the loss-in-volume ($V_0 - V$), where $V_0$ and $V$ are unperturbed and perturbed volumes respectively), so that we arrive at the following relationship:

$$\tilde{\Omega} - \Omega \approx \kappa(V_0 - V) \tag{13}$$

We validate model predictions of a correlation between regeneration flux ($\tilde{\Omega}$) and volume-loss between punctured and unpunctured ears ($V_0 - V$). Our experimental data (regeneration in blue curve in *Figure 4H* and *Figure 4—figure supplement 1E,F*) depicts a proportional relationship, thus validating the accuracy of our model. Our data shows that the unperturbed flux $\Omega \simeq 1 \mu m/hr$ and slope $\kappa$ is approximately 3–5.

## Modeling pressure-driven growth and deformation of the vesicle shape

The otic vesicle is expected to deform under the action of pressure, thus leading to growth and reshaping of the tissue. We therefore sought to assess how these forces affect individual cells. Given the vesicle geometry, cells in the otic vesicle experience three types of pressure-derived forces: (i) pressure force ($P$, $N.\mu m^{-2}$) normal to the apical membranes in a direction that flatten cells, (ii) tissue stress ($\sigma$, $N.\mu m^{-2}$) distributed normal to lateral membranes, and (iii) reactionary or support forces from hindbrain and skin tissue.

To formalize this, we modify a recent framework introduced to study the growth of cellular cysts (*Ruiz-Herrero et al., 2017*). In a closed geometry, tissue stress $\sigma$ is related to lumenal pressure $P$ by a simple force balance equation. For a spherical pressure vessel, the pressure force pushing one hemisphere is counterbalanced by tissue tension.

$$P\pi R^2 = 2\pi Rh\sigma \Longleftrightarrow \sigma = \frac{PR}{2h}. \tag{14}$$

To link tissue stress to deformation, we investigated the material properties of the tissue ($G$). We initially modeled otic tissue as being similar to a purely viscous fluid flowing under a tangential shear stress $\sigma$ with a strain-rate of $\dot{\epsilon} = \frac{1}{R}\frac{dR}{dT}$.

$$\sigma = G\dot{\epsilon}. \tag{15}$$

For a purely viscous fluid with viscosity coefficient $\mu$ ($Pa.h$) deforming in a spherically-restricted geometry, the shear stress ($\sigma$) in the fluid layers relates to rate of radius change using Stokes' law as:

$$\sigma = 4\mu \frac{1}{R}\frac{dR}{dt}. \tag{16}$$

Since a viscous material deforms continuously under the action of a force and retains the deformation when force is removed, we validated our assumption by studying the temporal trends in resting shapes of cells after pressure forces are eliminated. Indeed, we observed that the resting shape of cells (after puncture) had undergone irreversible deformation in developmental time (*Figure 3G and D* (red bars progressively decrease)). In other words, a cell at 35 hpf has a relatively squamous morphology at rest compared to the same cell at 25 hpf. When compared to the contralateral unpunctured ear, however, we observed an elastic response in punctured vesicles wherein cells had dynamically reshaped to more columnar morphologies (*Figure 3E and F,H* (difference between red and blue bars)). This suggested that the tissue behaved elastically in short time-scales and viscous in long time-scales. To incorporate elastic behavior on short timescales, we modeled the overall deformation as:

$$\dot{\sigma} + \frac{\sigma}{\tau} = G\dot{\epsilon} \tag{17}$$

We next quantified cell material properties by quantifying the short (elastic) and long timescale (viscous) responses in response to puncturing perturbations (*Figure 6A*). The reversible component of a cell's deformation, when all acting forces are eliminated, is inversely proportional to its elasticity ($K$). The irreversible component of a cell's deformation that accumulates over time is inversely proportional to the viscosity coefficient ($\mu$). We also investigated and identified the molecular origins of material property patterning in terms of actomyosin networks that were found to be apically expressed in the otic cells (*Figure 6*). Thus, our model identifies the role of pressure forces in deforming cells and driving growth. Material property patterning was found to locally modulate the deformation to anisotropically shape the vesicle through growth.

## Quantification and statistical analysis

Each data point of morphodynamic quantification was obtained using our automated bioimage informatics pipeline (ITIAT) on images from 10 different embryos that were immobilized with $\alpha$-bungarotoxin mRNA. 30 time-points were analyzed—every hour between 16 and 45 hpf—which included 300 otic vesicle lumen measurements and more than 200,000 segmented cells. For *Figure 1E–H*, *Figure 1—figure supplement 1N–O* the translucent spread of the data is the standard deviation.

For analysis of mitotic cell deformation, we identified 54 mitotic cells. The spread of the data presented as translucent overlays in *Figure 2G* is the calculated standard deviation. For measuring the hindbrain deformation in *Figure 2J*, n = 10 embryos/timepoint were used.

Pressure measurements were acquired from 5, 8, and nine different embryos at 30, 36, and 48 hpf (*Figure 3C*).

For puncture experiments, 10 otic vesicles were measured that were punctured and there morphometrics were compared to 10 unpunctured otic vesicles (*Figures 3E–H* and *4E–H*, *Figure 4—figure supplement 1*). The error bars are standard deviation and the p-values were obtained using a student t-test.

For ion-channel and pH perturbations, each data point is five embryos and error bars are standard deviation (*Figure 5*).

For calculating the effective tissue viscosity from *Equation 1*, the time-averaged data of all variables were obtained from the datasets shown in *Figure 3C,E–F*. The instantaneous growth rate, $R' = \mathrm{d}R/\mathrm{d}t$, was calculated from the difference of radius $R$ of each time point before averaging. A formula for error propagation (*Ku, 1966*) was derived from *Equation 1*:

$$\delta_\mu \mu = 18\sqrt{(\delta_P P)^2 + 2(\delta_R R)^2 + (\delta_h h)^2 + (\delta_{R'} R')^2},$$

where $\delta_X$ is the standard deviation of variable $X$. The mean and standard deviations are summarized in the table below:

| | | | Mean (Standard deviation) | | | |
|---|---|---|---|---|---|
| $t$ | $P$ | $R$ | $h$ | $R'$ | $\mu$ |
| hpf | Pa | μm | μm | $10^{-4}$ μm/s | $10^6$ Pa·s |
| 24-36 | 116 (23.1) | 36 (0.18) | 13 (0.12) | 2.31 (0.73) | 6.26 (0.30) |
| 36-48 | 166 (23.4) | 48 (0.22) | 10 (0.21) | 2.15 (0.95) | 22.2 (1.29) |

For the derivation of relative differences in tissue elasticity and viscosity, each data point was obtained from 10 different punctured embryos (*Figure 6A–C*) and the translucent spread represents the standard deviation.

*Figure 6I-J*, 22 utrophin expressing cells and 22 myosin expressing cells were analyzed. *Figure 6K–L–M*, scatterplots with 15, 16, and 12 embryos were used respectively.

## Data and software availability

### Image analysis

Our entire bioimage informatics pipeline called '*In toto image analysis toolkit (ITIAT)*' is available online at: https://wiki.med.harvard.edu/SysBio/Megason/GoFigureImageAnalysis. The pipeline can be used for generating 3D surface reconstructions, automatic whole-cell and nuclei segmentations, and cell tracking. The code is open-source and written using the C++ programming language. The code can be downloaded and compiled in any platform using CMake, a tool for generating native Makefiles. The code is built by linking to pre-compiled open-source libraries, namely, ITK (http://www.itk.org) (*Yoo et al., 2002*) and VTK toolkit (http://www.vtk.org) (*Schroeder et al., 2006*) for image analysis and visualization respectively. We also used the open-source and cross-platform GoFigure2 application software for the visualization, interaction, and semi-automated analysis of $3D + t$ image data (http://www.gofigure2.org) (Gelas, Mosaliganti, and Megason et al., in preparation). Measurement of mitotic cell aspect-ratios was carried out using ZEN (Carl Zeiss) software 3D distance functionality. Measurements were analysed and plotted with Matlab (Mathworks) and Microsoft Excel. To obtain 3D models of the otic vesicles, 2D contours were first placed along regularly-sampled z-planes in GoFigure2 (*Figure 1I*). 3D reconstructions were obtained using the Powercrust reconstruction algorithm (https://github.com/krm15/Powercrust) (*Amenta et al., 2001*). Automatic cell and lumen-segmentation were performed using ACME software for whole-cell segmentations (*Mosaliganti et al., 2012*; *Xiong et al., 2013*) (*Figure 1—figure supplement 1J–M*).

## Acknowledgements

We thank members of Megason lab for feedback, Mr. Dante D'India for fish care, and Ms. Suzanne Mosaliganti for help in proof-reading the manuscript. This work was supported by the National Institute of Health grant K25 HD071969 (KRM), Novartis Fellowship for Systems Biology (IAS), National Institute of Health grant 5F32HL097599 (IAS), A*STAR International Fellowship (CUC), Hearing Health Foundation (IAS), the MacArthur Foundation (LM), National Science Foundation grant BMMB 15–36616 (LM), National Institute of Health grant R01 DC010791 (SGM), and National Institute of Health Grant R01 DC015478 (SGM).

## Additional information

### Funding

| Funder | Grant reference number | Author |
|---|---|---|
| National Institutes of Health | K25 HD071969 | Kishore R Mosaliganti |
| Hearing Health Foundation | | Ian A Swinburne |
| National Institutes of Health | 5F32HL097599 | Ian A Swinburne |
| National Institutes of Health | DC010791 | Sean G Megason |
| National Institutes of Health | DC015478 | Sean G Megason |

| | | |
|---|---|---|
| John D. and Catherine T. MacArthur Foundation | | L Mahadevan |
| National Science Foundation | BMMB 15-36616 | L Mahadevan |
| Agency for Science, Technology and Research | | Chon U Chan |

The funders had no role in study design, data collection and interpretation, or the decision to submit the work for publication.

## Author contributions

Kishore R Mosaliganti, Conceptualization, Software, Formal analysis, Investigation, Writing—original draft, Writing—review and editing; Ian A Swinburne, Investigation, Writing—original draft, Writing—review and editing; Chon U Chan, Formal analysis, Investigation, Methodology, Writing—original draft, Writing—review and editing, Designed and characterized the pressure probe; Nikolaus D Obholzer, Amelia A Green, Shreyas Tanksale, Investigation, Writing—review and editing; L Mahadevan, Conceptualization, Formal analysis, Supervision, Writing—original draft, Writing—review and editing, Designed and characterized the pressure probe; Sean G Megason, Conceptualization, Formal analysis, Supervision, Funding acquisition, Writing—original draft, Project administration, Writing—review and editing

## Author ORCIDs

Ian A Swinburne https://orcid.org/0000-0003-4162-0508
Chon U Chan https://orcid.org/0000-0002-9047-057X
L Mahadevan https://orcid.org/0000-0002-5114-0519
Sean G Megason https://orcid.org/0000-0002-9330-2934

## Ethics

Animal experimentation: This study was performed in strict accordance with the recommendations in the Guide for the Care and Use of Laboratory Animals of the National Institutes of Health. The Harvard Medical Area Standing Committee on Animals approved zebrafish work under protocol number 04487.

## Decision letter and Author response

Decision letter https://doi.org/10.7554/eLife.39596.023
Author response https://doi.org/10.7554/eLife.39596.024

# Additional files

## Supplementary files

• Transparent reporting form
DOI: https://doi.org/10.7554/eLife.39596.020

## Data availability

All data generated or analyzed during this study are included in the manuscript and supporting files.

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
