## [Decision Letter]

[**Editorial note:** This article has been through an editorial process in which the authors decide how to respond to the issues raised during peer review. The Reviewing Editor's assessment is that all the issues have been addressed.]

Thank you for submitting your article "Size control of the inner ear via hydraulic feedback" for consideration by *eLife*. Your article has been reviewed by three peer reviewers, one of whom is a member of our Board of Reviewing Editors, and the evaluation has been overseen by Didier Stainier as the Senior Editor. The following individual involved in review of your submission has agreed to reveal her identity: Ksenia Gnedeva (Reviewer #3).

The Reviewing Editor has highlighted the concerns that require revision and/or responses, and we have included the separate reviews below for your consideration. If you have any questions, please do not hesitate to contact us.

Summary:

This paper examines the potential role of pressure in regulating the volume of the otic vesicle during zebrafish development. The authors present several lines of evidence that there is indeed transepithelial pressure: the shapes of adjacent epithelial cells are perturbed, the vesicle deflates when punctured and, using a clever device, they present some direct measurements of transepithelial pressures (values on the order of 100 Pa). They calculate, based on the geometry, the flux of fluid into the vesicle, which is fairly constant over the developmental times that they are studying. After puncturing the vesicle, the luminal volume "catches up" through an increase in the flux; they conclude that the pressure is inhibiting the flux, establishing a feedback loop.

Major concerns:

The reviewers have noted that the pressure measurements themselves are very noisy, and their reproducibility is unclear. These features make it difficult to judge the extent to which the very plausible theoretical model is or is not quantitatively consistent with the measurements. They have requested clarification on a number of aspects of the pressure measurements themselves, and asked if the connection between the theory and data can be made more precise than the mostly correlative comparison described in the paper.

Additionally, the reviewers have suggested that further analysis is warranted in order to solidify the claim that there is a causative connection between changes in hydrostatic pressure and changes in the rate of cell growth, cell proliferation, and/or cell cycle length. They have also asked about the connection between tissue-level tension and cell proliferation, the role of elasticity of the surrounding tissues, and the possibility of measuring that elasticity directly.

Separate reviews (please respond to each point):

*Reviewer #1:*

This is a very interesting paper on the problem of organ size regulation. It combines an impressive set of imaging and in vivo measurements with a simple mathematical model to paint a clear picture of a plausible mechanism at work the zebrafish. The logic of the analysis is very clear, as is the presentation.

Having said the above, I find that the manuscript falls short on two major fronts. First, if Figure 3—figure supplement 1B-D are a general indication of the repeatability of the pressure measurements in vivo, then it is not clear to this reviewer the extent to which truly quantitative conclusions about the system can be inferred. The authors have not included any discussion about the potential sources of the large fluctuations seen in these experiments and their variability from one experiment to the next. The case for publication in *eLife* would be substantially enhanced if the reliability of the measurements were quantified and understood more deeply.

A second major issue concerns the connection between the eminently plausible theory presented in the paper and the quantitative observations. The verbal discussion in the paper regarding a whole range of qualitative observations dovetails nicely with what would be expected from the theory, but yet the paper does not appear to have a single graph in which a quantitative prediction of the theory is compared against experiment. The case for publication in *eLife* would be much stronger if such a comparison were possible. Can the authors show that there is more than a qualitative relationship between the two?

Minor Comments:

As a minor comment I would urge the authors to move the essential aspects of the presentation of the theory into the Results section. The mathematics is very straightforward and it would increase readability to have it interspersed within the Results section where appropriate.

*Reviewer #2:*

This paper examines the potential role of pressure in regulating the volume of the otic vesicle during the zebrafish development. The authors present several lines of evidence that there is indeed transepithelial pressure: the shapes of adjacent epithelial cells are perturbed, the vesicle deflates when punctured and, using a clever device, they present some direct measurements of transepithelial pressures (values on the order of 100 Pa). They calculate, based on the geometry, the flux of fluid into the vesicle, which is fairly constant over the developmental times that they are studying. After puncturing the vesicle, the luminal volume "catches up" through an increase in the flux; they conclude that the pressure is inhibiting the flux, establishing a feedback loop.

I think that the most interesting aspect of the manuscript is the pressure measurements. All the rest is indirect and based on correlations: the pressure feedback model is a plausible explanation for the observed morphological changes. Without the pressure measurements, this is not be an *eLife* paper.

However, the pressure measurements themselves are poorly developed and not convincing. There are many technical aspects that are not addressed, and many controls that need to be done.

1) Are the pressure measurements themselves dependent on the ionic composition and/or electrical potential? This is potentially important as the endolymph has an unusual ionic concentration and voltage, at least in adults. Is the high potassium concentration established at these early times?

2) Does the pressure measurement depend on the diameter of the capillary. I am surprised and concerned by the wide range of diameters (2-20 microns).

i) Is there a problem with damage – does the capillary tend to puncture the vesicle like the puncture needle. If not, why not? No details are given about the puncture needle: an "unclipped glass needle", whatever that is. What is the diameter?

ii) Is there a problem with mixing of the deionized water in the capillary with the endolymph? I would expect the diffusive exchange of fluid to be large, especially with 20 micron diameter capillaries. It looks like the total volume of the capillary is much larger than the vesicle, so the endolymph will be greatly diluted, unless the ion pumps can keep up. How long is mixing expected to take.

3) There simply are not enough measurements of pressure to be convincing, given the variabilities of the individual measurements in Figure 3—figure supplement 1.

4) Figure 3—figure supplement 1 shows only a small part of the time traces. What is happening before time zero. For me to be convinced, I would like to see the whole time course starting with the capillary in the (undefined) Dannieau (sic) buffer, then the penetration of the skin, followed by the epithelium and finally entry into the vesicle lumen. How often does this experiment work? Have the recordings in Figure 3—figure supplement 1 been cherry-picked?

5) A key experiment will be to measure the pressure while puncturing and showing that it goes down (and perhaps recovers).

In summary. The most novel aspect of this paper is the pressure measurement. However, this is much too preliminary to be included in the current work. I suggest taking them out and submitting the manuscript to another journal. I do not believe that such a manuscript meets the level of conceptual advance for an *eLife* paper: the arguments are correlative and the model is only at the level of plausibility. In other words, the results in this manuscript are suggestive of a pressure-flux feedback model, which therefore remains a hypothesis. I encourage the authors to develop the pressure measurements, but there is a long way to go.

*Reviewer #3:*

The manuscript by Mosaliganti et al. investigates the mechanism of organ size control in the developing otic vesicle in zebrafish. In particular, the authors demonstrate that fluid transport into the otic vesicle results in stress and stretching of the otic epithelium, and claim that constitutes the major driving force in lumen expansion and organ growth. The authors also suggest that buildup of the pressure in the otocyst inhibits further fluid transport, thus repressing growth. Such a negative feedback mechanism can explain the observed robustness of the otic vesicle growth, and explain a catch-up growth phenomenon that they observe after puncturing the otocyst.

The authors characterize the vesicle's size using unbiased quantitative measurements, including direct measurements of the luminal pressure. Ouabain and morpholino experiments convincingly demonstrate the involvement of Na+-K^+^-Cl transporter, Slc12a2, in the fluid flux into the vesicle. Overall, the article is well written and beautifully illustrated. The experiments are easy to follow. All the numerical comparisons are accompanied by adequate statistical analysis. However, I believe the following concerns should be addressed in order to support the proposed conclusions.

Major concerns:

I do not entirely agree with the author's definition of growth. Organ growth is determined by an increase in tissue mass (via increase in cell number, cell size, or both). To be precise, otic vesicle growth, characterized in Figure 1F, is represented by the green curve (otic tissue volume), and not the blue curve (otic vesicle volume). Increase in the vesicle's lumen, especially after the puncture experiments, should not be called growth, if tissue volume does not change in these experiments. I, however, agree with the overall idea that hydrostatic pressure can be a driving force for tissue growth, as it may cause tissue stretching and cell proliferation, however, this is not characterized in the manuscript. In this regard, the effect of hydrostatic pressure on the rate of cell growth and proliferation, not on the changes in lumen volume, should be characterized. This should be done for all their experiments, especially for normal development and Slc12a2 morpholino experiments. It will be critical to demonstrate that there is a causative connection between changes in hydrostatic pressure, and changes in the rate of cell growth, cell proliferation, and/or cell cycle length. These data are already present in the time-lapse videos provided by the authors, and should be extracted/analyzed, as it is critical to link hydrostatic pressure to organ growth control mechanisms discussed by the authors in the Introduction of the manuscript.

– One of the concerns I have with the manuscript is that the authors seem to conclude that growth rate in the organ is determined by an equilibrium between hydrostatic pressure, regulated by fluid flux into the vesicle, and tension in the otic epithelium. Biologically, such an assumption is unlikely, as an increase in epithelial tension is a well-described trigger for cell proliferation and growth. Gudipaty et al., 2017 and Elosegui-Artola et al., 2017 are just a couple of recent articles on the matter. Authors do not consider the elastic forces exerted by the tissues surrounding the otic vesicle (e.g. mesenchyme, brain, and skin). In addition to the otic tissue viscoelastic properties, these forces are very likely to counteract the hydrolytic pressure in the vesicle, and to affect its growth. In fact, such a mechanism of growth control was recently shown to take place in the vestibular sensory organ in mice. This should be accounted for in the theoretical model proposed by the authors and mentioned in the discussion.

– Related to the previous comment, the authors estimate effective viscosity of the otic vesicle tissue from their theoretical model. To test this prediction, and to confirm the accuracy of the model, Young's modulus of the tissue should be measured directly (e.g via atomic force microscopy). Also in this context, the authors claim that puncturing perturbations (to eliminate pressure forces) allowed them to measure spatial differences in cell viscoelasticity. These experiments only suggested the local differences in tissue viscosity, direct measurements of cell stiffness should be done to confirm this suggestion.

– It is not apparent to me that the authors demonstrate clearly the causality of spatial differences in cell viscoelasticity. Again, although hydrostatic pressure sensed by all the cells in the vesicle should be the same, the local difference in the opposing elastic forces from surrounding tissues can create differential thinning of the vesicle's wall; this in turn may result in actomyosin skeleton remodeling and stiffness changes, rather than vice versa.

– In the section "tissue material properties are patterned through actomyosin regulation" the authors talk about WT and transgenic animals, without explaining what transgenes are used. It made it hard to review this section.

I'd like to disclose that as a biologist with limited biophysics background, I cannot fully review the mathematical aspects of the work, and trust that one of the other reviewers has the expertise necessary to judge the accuracy of the theoretical model proposed here.

Minor Comments:

– To test their model's predictions, the authors conduct puncture experiments to evaluate the role of hydrolytic pressure on the growth rate of the otic vesicle. Although the authors demonstrate that punctures heal fast relative to the growth rate of the organ, such manipulations are invasive and are not fully sufficient to support their conclusions. Direct manipulation of the chemi-osmotic potential, for example, could be done to change the rate of endolymph accumulation, and to test its effect on organ growth. This could be achieved by injecting hydrophilic inert compounds, such as ficoll, into the otic vesicle or into the tissue surrounding it. If water transport across an epithelium is, in fact, the main driving force for otic vesicle growth, increasing osmolality of the endolymph should induce accelerated growth and increase cell proliferation rates.

[Editors' note: further revisions were requested prior to acceptance, as described below.]

Thank you for resubmitting your work entitled "Size control of the inner ear via hydraulic feedback" for further consideration at *eLife*. Your revised article has been favorably evaluated by Didier Stainier (Senior Editor), a Reviewing Editor, and two of the original reviewers.

The reviewers are of the opinion that the manuscript has been improved but there are some remaining issues that need to be addressed before acceptance, as outlined below:

i) In the interests of clarity, we would like to see the pressure traces in the manuscript proper rather than in the supplementary figures.

ii) If indeed there is no direct measurement of the pressure drop during puncture this should be stated explicitly.

Other points:

1) Abstract: what is meant by "noise in underlying molecular and cellular processes"? This premise is not established in the paper. It is a straw man. How much expression noise would it take to mess up otic development?

2) Results: SD or SE? what is n?

3) Figure 1E: start the y-axes at zero (zero is important in this case)

4) Figure 1F: match the colors to Figures 1A-D

5) Figure 5: the Na,K-ATPase is generally referred to as a pump rather than an ion channel.

---

## [Author Response]

Major concerns:The reviewers have noted that the pressure measurements themselves are very noisy, and their reproducibility is unclear. These features make it difficult to judge the extent to which the very plausible theoretical model is or is not quantitatively consistent with the measurements. They have requested clarification on a number of aspects of the pressure measurements themselves, and asked if the connection between the theory and data can be made more precise than the mostly correlative comparison described in the paper.

We have performed additional pressure measurements, controls and clarifications to address the concerns, which are detailed below.

Our theoretical model has two aspects: 1. By accounting for geometry and conservation laws, a mechanical feedback mechanism emerges that regulates endolymph flux and organ size. 2. Once the mechanical equilibrium is reached, the epithelial tissue is further remodeled *at a longer time scale* determined by an effective tissue viscosity.

The first aspect is verified by successfully predicting the catch-up growth dynamics after puncturing experiments (see ‘Model prediction and validation: Pressure negatively regulates fluid flux’). Notably, the experimental data (Figure 4H and Figure 4—figure supplement 1E,F) can be fitted with the prediction (equation 13) with a proportional relationship related to the permeability coefficient *K.*

The second aspect is a general description of the force balance and the tissue’s viscoelastic material property. If the epithelial tissue viscosity can be independently measured, it would further support our pressure measurement, but not our theoretical framework. Nevertheless, we have considered measurement methods, such as pipette aspiration methods for living cells (Hochmuth, 200). However, the effective viscosity of the epithelial tissue, based on our estimation from pressure measurements, is 4-5 orders of magnitude higher than the cellular viscosity. Deploying the same tool on the epithelial tissue, it would require a much higher suction or a much longer waiting period. It is not possible to measure the viscosity of the in vivo tissue without perturbing its structural properties and shape.

In summary, the framework of our model has been verified with qualitative data. The effective tissue viscosity measurement agrees up to the order of magnitude with other systems. Independent viscosity measurement is not possible due to technical difficulty, yet this is not relevant to the basis of our model.

Hochmuth, R.M., 2000. Micropipette aspiration of living cells. *Journal of biomechanics, 33*(1), pp.15-22.

Additionally, the reviewers have suggested that further analysis is warranted in order to solidify the claim that there is a causative connection between changes in hydrostatic pressure and changes in the rate of cell growth, cell proliferation, and/or cell cycle length. They have also asked about the connection between tissue-level tension and cell proliferation, the role of elasticity of the surrounding tissues, and the possibility of measuring that elasticity directly.

May we clarify that there are no claims within the manuscript on causality between pressure and the rate of cell growth, proliferation, and cell cycle length. Pursuing these connections, while of great interest to us, is beyond the original scope of this study.

Separate reviews (please respond to each point):

Reviewer #1:

*This is a very interesting paper on the problem of organ size regulation. It combines an impressive set of imaging and* in vivo *measurements with a simple mathematical model to paint a clear picture of a plausible mechanism at work the zebrafish. The logic of the analysis is very clear, as is the presentation.*

We thank reviewer #1 on the positive feedback and critical review.

*Having said the above, I find that the manuscript falls short on two major fronts. First, if Figure 3—figure supplement 1B-D are a general indication of the repeatability of the pressure measurements* in vivo*, then it is not clear to this reviewer the extent to which truly quantitative conclusions about the system can be inferred. The authors have not included any discussion about the potential sources of the large fluctuations seen in these experiments and their variability from one experiment to the next. The case for publication in eLife would be substantially enhanced if the reliability of the measurements were quantified and understood more deeply.*

We have performed more characterization on the pressure probe (see Figure 3—figure supplement 1 and further discussion in reviewer #2). Also, more measurements were performed and the statistics are depicted in Figure 3C.

“Pressure measurements were acquired from 5, 8, and 9 different embryos at 30, 36, and 48 hpf (Figure 3C).”

“Otic vesicle pressures at different developmental stages of wild-type zebrafish embryos (red diamond: mean value. *p<5.0e-2).”

The additional measurements altered the mean pressure measurements used to approximate the viscosity of the tissue.

Adjusted calculations are detailed in the text:

“Using this relationship, our morphodynamic measurements, and pressure measurements we estimate the effective viscosity of the otic vesicle tissue to be about 6.3+/-0.30 x 10^6 Pa*s from 24-36 hpf and then 2.2+/-0.13 x 10^7 Pa*s from 36-48 hpf (see Materials and methods for error propagation calculations).”

“The mean and standard deviations are summarized in the table below”:

(see table in manuscript)

A discussion on the potential sources of the fluctuations is added:

“At the typical rise time (around 0.5 minute) of the probing stage, (Stage II in Figure 3—figure supplement 1F), the concentration remains at about 70%-90% for inner diameter of $15-5um. We expect the impact on the pressure reading was small at this time scale. After about 5 to 12 minutes, the concentration drops to 10%, which may significant modify the chemical potential. Together with the imperfection in sealing, they could contribute to the fluctuation measured at the longer time scale. However, we have ignored some factors that can maintain the ionic concentration: the active transport of ions and the potentially higher viscosity in the lumen.”

A second major issue concerns the connection between the eminently plausible theory presented in the paper and the quantitative observations. The verbal discussion in the paper regarding a whole range of qualitative observations dovetails nicely with what would be expected from the theory, but yet the paper does not appear to have a single graph in which a quantitative prediction of the theory is compared against experiment. The case for publication in eLife would be much stronger if such a comparison were possible. Can the authors show that there is more than a qualitative relationship between the two?

As discussed above, the most important prediction of a negative feedback mechanism (equation 11-13) is verified by obtaining a proportional relationship between regeneration flux and volume-loss between punctured and unpunctured ears (see ‘Modeling pressure generation and feedback to fluid transport mechanisms’). Eliminating this single fitting parameter requires an independent measurement of the permeability coefficient, which is not feasible.

Minor Comments:As a minor comment I would urge the authors to move the essential aspects of the presentation of the theory into the Results section. The mathematics is very straightforward and it would increase readability to have it interspersed within the Results section where appropriate.

We agree with the inclination to place the theory in front of the Results section. However, much of our audience (see reviewer #3) may be distracted by the theory. We have decided to keep the theory consolidated within the Materials and methods section.

Reviewer #2:

[…] I think that the most interesting aspect of the manuscript is the pressure measurements. All the rest is indirect and based on correlations: the pressure feedback model is a plausible explanation for the observed morphological changes. Without the pressure measurements, this is not be an eLife paper.However, the pressure measurements themselves are poorly developed and not convincing. There are many technical aspects that are not addressed, and many controls that need to be done.

We thank reviewer #2 for the suggestions on the pressure measurements, which help us to clarify the measurement technique to general readers.

1) Are the pressure measurements themselves dependent on the ionic composition and/or electrical potential? This is potentially important as the endolymph has an unusual ionic concentration and voltage, at least in adults. Is the high potassium concentration established at these early times?

We have added controls where calibrations were performed with distilled water or a solution that resembles mature endolymph. There is no dependence on the ionic composite regardless of the tip size. We have added Figure 3—figure supplement 1B and the following paragraph:

“Similar tests were conducted with various capillary diameters and ionic concentrations within the bath (deionized water and a solution that resembles mature endolymph) to ensure there is no additional effect (Figure 3—figure supplement 1B).”

In Caption of Figure 3—figure supplement 1E:

“Calculations of diffusive mixing between endolymph and capillary filling after puncturing. Their initial ionic concentration are C0 and 0, respectively. The mean ionic concentration inside the vesicle C decreases over time t at a rate depending on the capillary inner diameter d. (Inset) Selected solutions are shown for d=5,15um in the first 5 minutes.”

and:

“The composition of our synthetic endolymph is 5 mM sodium chloride, 150 mM potassium chloride, 0.2 mM calcium chloride, 0.5 mM glucose, 10 mM tris, buffered to pH 7.5.”

While it is unknown what the ionic composition is for the early zebrafish otic vesicle, it is most likely of high sodium content, like plasma, as it is within embryonic otic cysts of mammals.

2) Does the pressure measurement depend on the diameter of the capillary. I am surprised and concerned by the wide range of diameters (2-20 microns).

We now perform the pressure measurement on a microscope with higher resolution and the tip size are more accurately measured. We found that the inner diameter is between 6-13 microns and have changed the text accordingly.

“The sensor was coupled via a high pressure fitting to a 2 cm long glass capillary (World Precision Instruments) with a conical tip of 6-13 μm inner diameter (Figure 3A).”

To address the potential dependency on capillary diameter, we performed calibrations with tips of 4.5 and 10 microns inner diameter (as presented in comment #1 above) and found no dependence. Additionally, there is no dependence in our test data, as shown in Author response image 1:

i) Is there a problem with damage – does the capillary tend to puncture the vesicle like the puncture needle. If not, why not? No details are given about the puncture needle: an "unclipped glass needle", whatever that is. What is the diameter?

The puncture needles were similar sizes (5-10 microns) but wiggling is required to promote a larger wound upon removal. Insertion of the pressure probe was a much gentler and controlled motion to help a seal around the inserted glass capillary. Since the epithelium is under tension, we observe that it wraps around the tip once being punctured.

ii) Is there a problem with mixing of the deionized water in the capillary with the endolymph? I would expect the diffusive exchange of fluid to be large, especially with 20 micron diameter capillaries. It looks like the total volume of the capillary is much larger than the vesicle, so the endolymph will be greatly diluted, unless the ion pumps can keep up. How long is mixing expected to take.

To estimate the time scale at which the endolymph is diluted by the diffusive exchange, we numerically solve the diffusion equation with the vesicle-tip geometry and calculate the average ionic concentration inside the vesicle. As the inner diameter of the tip increases from 5 microns to 15 microns, the period before the endolymph drops to 10% decreases from 12 minutes to 5 minutes. As it typically takes less than one minute for the pressure to build up after the puncture, the diffusive exchange should not affect the plateau pressure reading. The subsequent exchange dilutes the endolymph and may be partly responsible to the pressure fluctuation. We added Figure 3—figure supplement 1E and the following paragraph:

“We also estimated the time scale at which the endolymph is diluted by the diffusive exchange. […] However, we have ignored some factors that can maintain the ionic concentration: the active transport of ions and the potentially higher viscosity in the lumen of the otic vesicle.”

3) There simply are not enough measurements of pressure to be convincing, given the variabilities of the individual measurements in Figure 3—figure supplement 1.

We have performed additional experiments and the statistics are shown in Figure 3C. The result is statistically significant to show that the pressure builds up with ages and therefore agrees with our prediction. A discussion on the potential source of fluctuation is added (see comment #2 of reviewer #1).

4) Figure 3—figure supplement 1 shows only a small part of the time traces. What is happening before time zero. For me to be convinced, I would like to see the whole time course starting with the capillary in the (undefined) Dannieau (sic) buffer, then the penetration of the skin, followed by the epithelium and finally entry into the vesicle lumen. How often does this experiment work? Have the recordings in Figure 3—figure supplement 1 been cherry-picked?

From the microscope, we observed that the capillary tip first indents the vesicle slightly before it punctures and being wrapped by the epithelium. Therefore, there is no distinguishable moment between skin and epithelium entry.

The criteria for accepting a measurement is the following: 1. The capillary is punctured at the correct location and depth. 2. The vesicle remains visually intact 3. The pressure builds up after the puncture and reaches a plateau. 4. After withdrawing the capillary the pressure drops rapidly to near the hydrostatic pressure at that depth, indicating the pressure is not built up from clogging. The overall successful rate of the puncturing attempts is about 10%. The traces of all data points in Figure 3C are shown in Figure 3—figure supplement 2.

Before time zero, the tip is placed beside the vesicle to measure the hydrostatic pressure at that depth. At that moment, the trace is the same as the calibration curve. We further clarify the measurement dynamic by adding Figure 3—figure supplement 1F:

“F. Stages in an otic vesicle pressure measurement. Upper: zooming into the first 1.5 minute. I: the tip was placed near the vesicle. The hydrostatic pressure was used as the baseline. II: after puncturing, the pressure built up gradually. III: after reaching a plateau, the pressure fluctuated around a mean value. This value was taken as the measurement result. IV: Upon withdrawing the tip from the vesicle, the pressure dropped to the base line, proving that the probe had been sensing the hydrostatic pressure in the enclosed domain.”

We have corrected the spelling errors and defined Danieau buffer.

“The composition of the Danieau buffer is 14.4 mM sodium chloride, 0.21 mM potassium chloride, 0.12 mM magnesium sulfate, 0.18 mM calcium nitrate, and 1.5 mM HEPES buffered to pH 7.6.”

5) A key experiment will be to measure the pressure while puncturing and showing that it goes down (and perhaps recovers).

Unlike an isolated cyst in which two capillaries can be punctured from both sides, it is challenging to introduce another capillary to otic vesicle without perturbing the sealing. Particularly, a wiggling action is required to open up a wound (as discussed above). An alternative way to confirm the fact that we are sensing the hydrostatic pressure, as we applied to all our tests, is to monitor the rapid pressure drop after withdrawing from the closed domain between the sensor and lumen (Stage IV Figure 3—figure supplement 1F).

In summary. The most novel aspect of this paper is the pressure measurement. However, this is much too preliminary to be included in the current work. I suggest taking them out and submitting the manuscript to another journal. I do not believe that such a manuscript meets the level of conceptual advance for an eLife paper: the arguments are correlative and the model is only at the level of plausibility. In other words, the results in this manuscript are suggestive of a pressure-flux feedback model, which therefore remains a hypothesis. I encourage the authors to develop the pressure measurements, but there is a long way to go.

While we appreciate the recognition of the novelty, we believe that there are additional features of the manuscript that are novel and of interest to the *eLife* audience. For instance, the systematic analysis of the organ growth, as summarized in Figure 1, is a more complete picture of an organ’s early growth than those previously presented. Additionally, the finding and quantification of a novel instance of catch-up growth are of general interest (Figures 3 and 4). Also, the theoretical model that accounts for geometry and conversation laws is self-consistent with various quantitative observations. Although the result may appear to be just correlative, they are supported by physical principles, quantification data, and experiments, which is beyond a hypothesis.

Reviewer #3:

[…] The authors characterize the vesicle's size using unbiased quantitative measurements, including direct measurements of the luminal pressure. Ouabain and morpholino experiments convincingly demonstrate the involvement of Na+-K^+^-Cl transporter, Slc12a2, in the fluid flux into the vesicle. Overall, the article is well written and beautifully illustrated. The experiments are easy to follow. All the numerical comparisons are accompanied by adequate statistical analysis. However, I believe the following concerns should be addressed in order to support the proposed conclusions.

We thank reviewer #3 for the positive feedback and insightful comments that help us to clarify the concepts.

Major concerns:I do not entirely agree with the author's definition of growth. Organ growth is determined by an increase in tissue mass (via increase in cell number, cell size, or both). To be precise, otic vesicle growth, characterized in Figure 1F, is represented by the green curve (otic tissue volume), and not the blue curve (otic vesicle volume). Increase in the vesicle's lumen, especially after the puncture experiments, should not be called growth, if tissue volume does not change in these experiments.

Although organ growth is commonly simplified to just an increase in cell number, we argue that this ignores a great amount of developmental biology and physiology which show how water, ECM, mineralization, and other factors besides cell proliferation are essential aspects of organs. While cell size and cell number contribute to organ growth, we think it is incorrect to dismiss the contribution of luminal volume. 70% of an average cell’s volume (and mass) is water, but one would never consider dismissing this tightly regulated volume as not being part of a cell’s size and function.

The composition and volume of the ear’s lumen, which contributes upwards of 50% of the organ’s volume, is as necessary to the organ’s function and physiology as the composition of the cytoplasm within its tissue’s cells. Because of this, there exist mechanisms in which it is regulated by the ear’s tissues. Additionally, lumen size and expansion are necessary during development to create space for sculpting the semicircular canals. The luminal volume and composition are as critical to the ear as extracellular collagen and hydroxyapatite are to a bone’s size and function (they contribute the majority of volume and mass to bones). To understand the growth of a bone, one should consider how the developing organ controls the deposition of these extracellular components. Likewise, to understand the growth of the ear and other fluid filled organs, it is essential to consider the lumen as part of the organ. We have added the following text to our manuscript to clarify this definitional point.

“Just as water is fundamental to the size and function of a cell's cytoplasm, the fluids filling the lumens of these organs, which are central to their development and physiological function, are fundamental components of these organs.”

I, however, agree with the overall idea that hydrostatic pressure can be a driving force for tissue growth, as it may cause tissue stretching and cell proliferation, however, this is not characterized in the manuscript. In this regard, the effect of hydrostatic pressure on the rate of cell growth and proliferation, not on the changes in lumen volume, should be characterized. This should be done for all their experiments, especially for normal development and Slc12a2 morpholino experiments. It will be critical to demonstrate that there is a causative connection between changes in hydrostatic pressure, and changes in the rate of cell growth, cell proliferation, and/or cell cycle length. These data are already present in the time-lapse videos provided by the authors, and should be extracted/analyzed, as it is critical to link hydrostatic pressure to organ growth control mechanisms discussed by the authors in the Introduction of the manuscript.

While we are interested in the molecular and cellular feedback response to pressure controlled growth, these avenues were beyond the original scope of this manuscript.

– One of the concerns I have with the manuscript is that the authors seem to conclude that growth rate in the organ is determined by an equilibrium between hydrostatic pressure, regulated by fluid flux into the vesicle, and tension in the otic epithelium. Biologically, such an assumption is unlikely, as an increase in epithelial tension is a well-described trigger for cell proliferation and growth. Gudipaty et al., 2017 and Elosegui-Artola et al., 2017 are just a couple of recent articles on the matter. Authors do not consider the elastic forces exerted by the tissues surrounding the otic vesicle (e.g. mesenchyme, brain, and skin). In addition to the otic tissue viscoelastic properties, these forces are very likely to counteract the hydrolytic pressure in the vesicle, and to affect its growth. In fact, such a mechanism of growth control was recently shown to take place in the vestibular sensory organ in mice. This should be accounted for in the theoretical model proposed by the authors and mentioned in the discussion.

Although we do not quantify the cellular feedback response to pressure, the cell proliferation and growth have been captured by the effective viscosity of the epithelial tissue (see equations 3, 7 and 16). We now point out that neighboring tissues likely contribute to the material properties of the otic vesicle’s tissue.

“Global constraints on the otic vesicle

The otic vesicle is not growing in isolation. In the embryo, it is immediately surrounded by extracellular matrix, mesenchymal cells, skin, and the brain. Within our model, these influences are abstracted as the effective material properties of the otic vesicle tissue. In fact, they may set limits to growth where the tension within the tissue begins to increase rapidly. We are likely observing an influence of these boundary conditions when we observe the spatial patterning of actinomyosin localization and regional tissue thinning (Figure 6). This boundary condition may accelerate cellular and molecular feedback mechanisms that were beyond the scope of this work. For instance, the cells within the tissue may respond to elevated tension by modulating proliferation rates, which may effectively alter the material properties of the tissue and alter strain (Halder and Johnson, 2011, Gudipaty et al., 2017, Gnedeva et al., 2017).”

The goal of the model was to relate fluid flux, hydrostatic pressure, material properties, and tension, geometry, and size to better understand organ growth. To simplify the model, we did not explicitly account for the molecular changes that can occur within cells in response to pressure to modulate its material properties by changing molecular adhesion states, cell division, or cell contraction. We now recognize the importance of this connection when we speculate on the underlying molecular mechanisms that modulate the mesoscopic features of interest.

– Related to the previous comment, the authors estimate effective viscosity of the otic vesicle tissue from their theoretical model. To test this prediction, and to confirm the accuracy of the model, Young's modulus of the tissue should be measured directly (e.g via atomic force microscopy). Also in this context, the authors claim that puncturing perturbations (to eliminate pressure forces) allowed them to measure spatial differences in cell viscoelasticity. These experiments only suggested the local differences in tissue viscosity, direct measurements of cell stiffness should be done to confirm this suggestion.

Young’s modulus is a measure of the material’s elastic properties, which dominates on short time-scales as seen during organ collapse when the vesicle is punctured. Viscosity, on the other hand, dominates on the longer timescales of organ growth. We measure relative elasticity and viscosity over time and space in Figure 6. An absolute measurement of Young’s modulus with AFM would either require isolating the otic vesicle tissue (which we tried but it is too fragile, and the properties would likely change) or if left in vivo the measurement would be a composite of all the tissues. We also think our theoretical model stands without this measurement.

– It is not apparent to me that the authors demonstrate clearly the causality of spatial differences in cell viscoelasticity. Again, although hydrostatic pressure sensed by all the cells in the vesicle should be the same, the local difference in the opposing elastic forces from surrounding tissues can create differential thinning of the vesicle's wall; this in turn may result in actomyosin skeleton remodeling and stiffness changes, rather than vice versa.

This is a great point and we have clarified our interpretation of the pattern we observed.

“As it is unclear what contribution the neighboring tissue has to the effective material properties of the growing otic vesicle, we are unable to distinguish whether the correlation between actomyosin patterns and tissue thinning is organ autonomous or whether elastic forces from neighboring tissue are influencing these behaviors.”

– In the section "tissue material properties are patterned through actomyosin regulation" the authors talk about WT and transgenic animals, without explaining what transgenes are used. It made it hard to review this section.

The transgenes were previously defined within the figure legend, Materials and methods, and Key Resource Table. We added labels to the figure panels and clarified the text within the Results, as follows:

“To identify how cell material properties are patterned, we examined localization patterns of F-actin and Myosin II using transgenic zebrafish (*Tg(actb2:myl12.1-eGFP)^e2212^* for visualizing myosin II distribution, and *Tg(actb2:GFP-Hsa.UTRN) ^e116^* for visualizing F-actin distribution (Behrndt et al., 2012).”

I'd like to disclose that as a biologist with limited biophysics background, I cannot fully review the mathematical aspects of the work, and trust that one of the other reviewers has the expertise necessary to judge the accuracy of the theoretical model proposed here.Minor Comments:– To test their model's predictions, the authors conduct puncture experiments to evaluate the role of hydrolytic pressure on the growth rate of the otic vesicle. Although the authors demonstrate that punctures heal fast relative to the growth rate of the organ, such manipulations are invasive and are not fully sufficient to support their conclusions. Direct manipulation of the chemi-osmotic potential, for example, could be done to change the rate of endolymph accumulation, and to test its effect on organ growth. This could be achieved by injecting hydrophilic inert compounds, such as ficoll, into the otic vesicle or into the tissue surrounding it. If water transport across an epithelium is, in fact, the main driving force for otic vesicle growth, increasing osmolality of the endolymph should induce accelerated growth and increase cell proliferation rates.

The otic vesicle’s lumen is both a small volume (~0.2 nL) and pressurized. We are unable to inject a consistent small volume (~0.01 nL) in a manner where the tissue was not perturbed. We are pursuing better ways to understand the molecular and cellular feedback response to pressure controlled growth, but these avenues were beyond the scope of this manuscript.

Our addition:

Since our initial submission, an excellent paper studying bone catch-up growth and size control was published. We have updated the Introduction to recognize this work.

“Recently, the related phenomenon of organ symmetry has been addressed in the context of tails and the inner ear; but, the control mechanism underlying catch-growth was not clearly identified (Rosello-Diez, Stephen and Joyner, 2017; Das et al., 2017, Green et al., 2017). Catch-up growth also occurs during bone growth and its study has clarified insulin signaling activity as being important for bone size control (Rosello-Diez and Joyner et al., 2015, Rosello-Diez et al., 2018). Nonetheless, catch-up growth has been underused in the study of vertebrate specific mechanisms of organ size control (Rosello-Diez et al., 2018).”

[Editors' note: further revisions were requested prior to acceptance, as described below.]

The reviewers are of the opinion that the manuscript has been improved but there are some remaining issues that need to be addressed before acceptance, as outlined below:i) In the interests of clarity, we would like to see the pressure traces in the manuscript proper rather than in the supplementary figures.

The pressure traces have been moved to Figure 3D. Figure references within text have been changed accordingly.

ii) If indeed there is no direct measurement of the pressure drop during puncture this should be stated explicitly.

We do not know whether there is a pressure drop upon insertion of the probe into the otic vesicle and have stated this explicitly.

“We are uncertain whether there is a drop in pressure upon insertion of the pressure probe into the otic vesicle because there is no alternate measurement device.”

Other points:1) Abstract: what is meant by "noise in underlying molecular and cellular processes"? This premise is not established in the paper. It is a straw man. How much expression noise would it take to mess up otic development?

We have changed the Abstract to the following:

“Animals make organs of precise size, shape, and symmetry. How developing embryos consistently make organs is largely unknown.”

2) Results: SD or SE? what is n?

“for all data-points in Figure 1 n=10 otic vesicles, data spread is the standard deviation)”

3) Figure 1E: start the y-axes at zero (zero is important in this case)

The y-axes now begins at zero.

4) Figure 1F: match the colors to Figures 1A-D

The colors now match Figures 1A-D.

5) Figure 5: the Na,K-ATPase is generally referred to as a pump rather than an ion channel.

The title in Figure 5 is now:

“Figure 5: Ear size is affected by disruptions in ion transport.”